# Electrophysiology and morphology of human cortical supragranular pyramidal cells in a wide age range

Pál Barzó[1†], Ildikó Szöts[2†], Martin Tóth[2], Éva Adrienn Csajbók[2], Gábor Molnár[2*‡], Gábor Tamás[2*‡]

[1]Department of Neurosurgery, University of Szeged, Szeged, Hungary; [2]HUN-REN-SZTE Research Group for Cortical Microcircuits, Department of Physiology, Anatomy and Neuroscience, University of Szeged, Szeged, Hungary

**\*For correspondence:**
molnarg@bio.u-szeged.hu (GM);
gtamas@bio.u-szeged.hu (GT)

[†]These authors contributed equally to this work
[‡]These authors also contributed equally to this work

**Competing interest:** The authors declare that no competing interests exist.

## eLife Assessment

In this revised work, Barzó et al. assessed the electrophysiological and anatomical properties of a large number of layer 2/3 pyramidal neurons in brain slices of human neocortex across a wide range of ages, from infancy to elderly individuals, using whole-cell patch clamp recordings and anatomical reconstructions. This large data set represents an **important** contribution to our understanding of how these properties change across the human lifespan, supported by **convincing** data and analyses. The authors have addressed the concerns raised in previous reviews. Overall, this study strengthens our understanding of how the neural properties of human cortical neurons change with age and will contribute to building more realistic models of human cortical function.

**Abstract** The basic excitatory neurons of the cerebral cortex, the pyramidal cells, are the most important signal integrators for the local circuit. They have quite characteristic morphological and electrophysiological properties that are known to be largely constant with age in the young and adult cortex. However, the brain undergoes several dynamic changes throughout life, such as in the phases of early development and cognitive decline in the aging brain. We set out to search for intrinsic cellular changes in supragranular pyramidal cells across a broad age range: from birth to 85 y of age and we found differences in several biophysical properties between defined age groups. During the first year of life, subthreshold and suprathreshold electrophysiological properties changed in a way that shows that pyramidal cells become less excitable with maturation, but also become temporarily more precise. According to our findings, the morphological features of the three-dimensional reconstructions from different life stages showed consistent morphological properties and systematic dendritic spine analysis of an infantile and an old pyramidal cell showed clear significant differences in the distribution of spine shapes. Overall, the changes that occur during development and aging may have lasting effects on the properties of pyramidal cells in the cerebral cortex. Understanding these changes is important to unravel the complex mechanisms underlying brain development, cognition, and age-related neurodegenerative diseases.

## Introduction

After birth, the brain undergoes developmental changes for a prolonged time that involve a series of complex and accurately orchestrated processes (*Rakic, 2009*). The production and migration of neurons is largely complete at the beginning of postnatal development, and then the intrauterine developmental processes continue: gray and white matter thickening, myelination, synaptogenesis,

pruning, and establishment of the basic anatomical architecture for initial neural pathway function. Subsequently, local connections within cortical circuits are fine-tuned, and increasingly complex, longer-term connections are established between circuits (*Stiles and Jernigan, 2010*). After that, the changes do not end, but continue throughout human life. They are mostly driven by environmental influences and experiences and lead to changes in metabolic activities (*Kuzawa et al., 2014*), changes in functional connectivity patterns (*Kelly et al., 2009*) and, with the maturation of white matter (*Beck et al., 2021*; *Yeatman et al., 2014*) changes in the speed of long-distance transmission (*van Blooijs et al., 2023*). The final phase is aging, where it slowly declines with advancing age, leading to a decline in cognitive signal processing functions and often resulting in neurodegenerative diseases (*Peters, 2006*). The cortical supragranular glutamatergic cell (or pyramidal cell) provides the excitatory synaptic inputs for local inhibitory circuitry and other pyramidal cells by which they create distinct subnetworks (*Yoshimura et al., 2005*). The development and formation of dendrites (*Petanjek et al., 2008*; *Koenderink and Uylings, 1995*) and synapses (*Huttenlocher and Dabholkar, 1997*) of pyramidal cells in the human cerebral cortex has been documented to some extent by postmortem studies, besides much less is known about their biophysical maturation and electrical properties in the early stages of development and the subsequent change or maintenance in later ages. Numerous studies demonstrated in non-primate animal models that the electrical characteristics of neurons change prominently in the early postnatal stage (*Molnár et al., 2020*). Changes in the intrinsic membrane properties (*Picken Bahrey and Moody, 2003*), the input resistance or the kinetics of the elicited action potentials were reported (*Kroon et al., 2019*; *Elston and Fujita, 2014*) in connection with maturation of macaque and rodent pyramidal cells. To date, however, no cross-age studies have been conducted on the electrophysiological parameters of human pyramidal neurons.

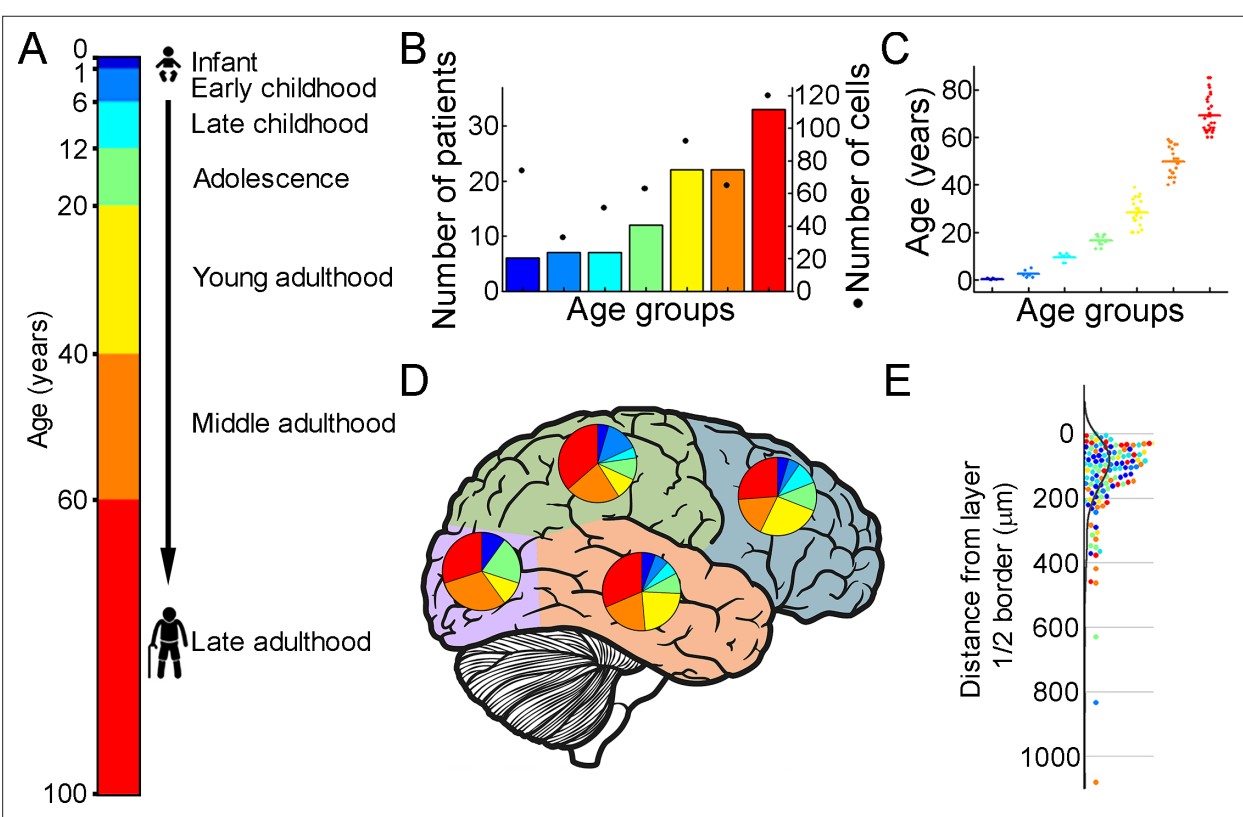

**Figure 1.** Illustration of the patient data on the samples utilized. (**A**) Illustration of the defined age groups. (**B**) Number of patients involved in age groups, (n=6, 7, 7, 12, 22, 22, 33 from infant to late adulthood, respectively). Dots show the number of human layer 2/3 pyramidal cells in our dataset regarding the defined age groups (n=74, 33, 51, 63, 92, 66, 120 from infant to late adulthood, respectively). (**C**) Distributions of patient ages within age groups. (**D**) Brain model indicates the number of surgically removed tissues from the cortical lobes. Colors indicate age groups. (**E**) The distribution of recovered cell bodies distance from the L1/2 border.

The online version of this article includes the following figure supplement(s) for figure 1:

**Figure supplement 1.** Patient metadata.

We have studied in detail the postnatal lifetime profile of the physiological and morphological properties of supragranular (layer 2/3) neurons of human pyramidal cells from neurosurgical resections. To this end, we performed whole-cell patch-clamp recordings and 3D anatomical reconstructions of human cortical pyramidal cells from 109 patients aged 1 m to 85 y for comprehensive data analysis to obtain the morphoelectric lifetime profile of supragranular pyramidal cells.

## Results

### Age-dependent differences in intrinsic subthreshold membrane properties

To extract biophysical properties of excitatory cells of human brain specimens we performed whole-cell patch-clamp recordings of pyramidal cells from neurosurgically removed human neocortical tissue sections. The samples were mainly from the frontal and temporal lobes (*Figure 1D*, *Figure 1—figure supplement 1D*), mostly from patients with tumors or hydrocephalus (*Figure 1—figure supplement 1B*, *Supplementary file 1*). Data were collected from 109 patients aged 1 m to 85 y (*Figure 1B and C*) from 498 human cortical layer 2/3 (L2/3) pyramidal cells. To confirm that the studied pyramidal cells

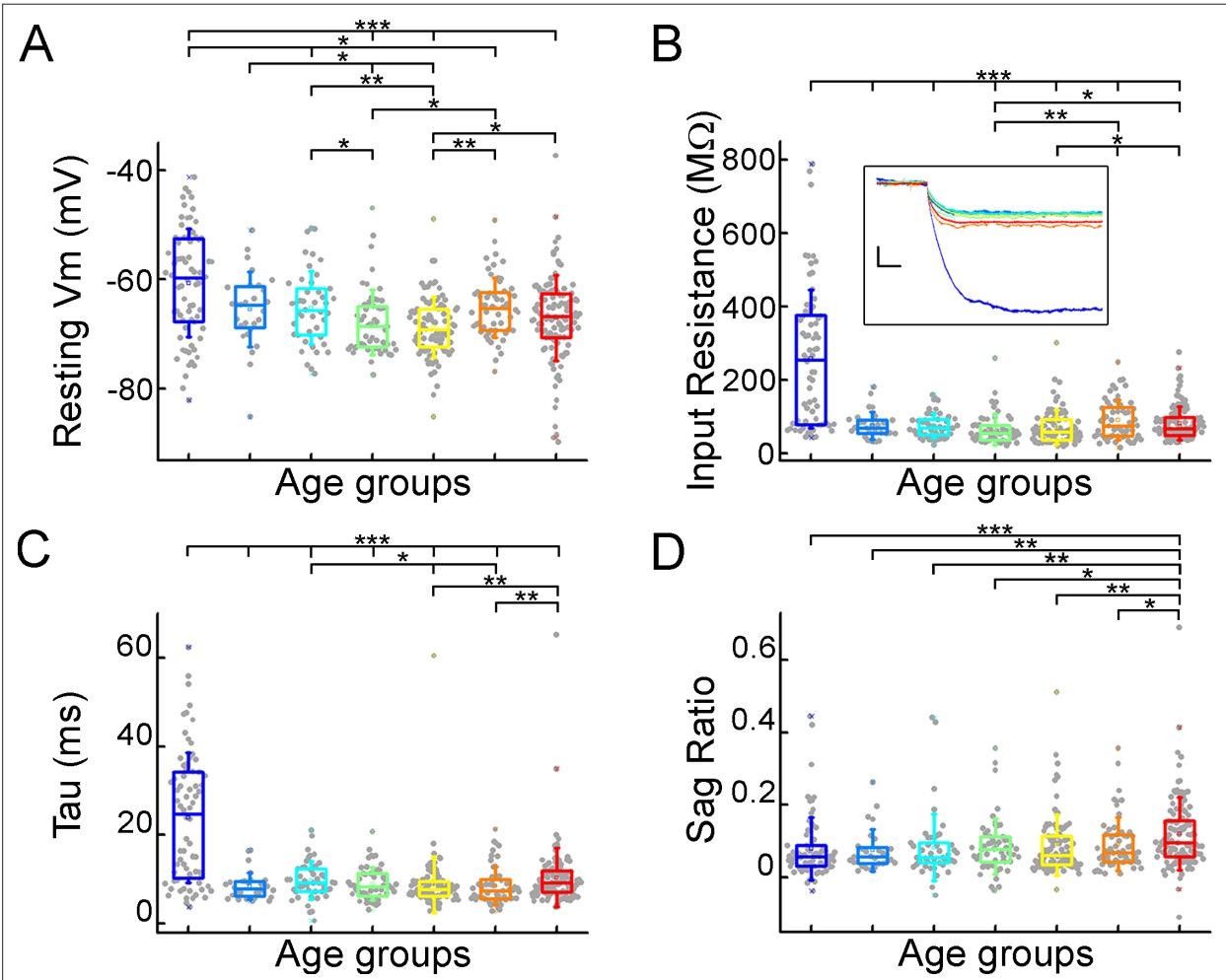

**Figure 2.** Subthreshold membrane properties vary across life stage. (**A–D**) Boxplots show resting membrane potential (**A**), input resistance (**B**), tau (**C**), and sag ratio (**D**) distributions in various age groups. Inset shows representative voltage traces from each group. Scale bar: 5 mV; 20 ms. Asterisks indicate significance (Kruskal–Wallis test with post-hoc Dunn test, *p<0.05, **p<0.01, ***p<0.001).

The online version of this article includes the following figure supplement(s) for figure 2:

**Figure supplement 1.** Distribution of subthreshold electrophysiological features with age.

**Figure supplement 2.** Electrophysiological differences during the first year of life.

**Table 1.** Subthreshold membrane properties.

| Subthreshold properties | Infant N=72 | Early childhood N=28 | Late childhood N=45 | Adolescence N=54 | Young adulthood N=89 | Middle adulthood N=56 | Late Adulthood N=113 |
|---|---|---|---|---|---|---|---|
| | Mean ± SD | Mean ± SD | Mean ± SD | Mean ± SD | Mean ± SD | Mean ± SD | Mean ± SD |
| Resting Vm (mV) | –60.64±9.86 | –65.44±6.82 | –65.17±6.68 | –67.86±5.94 | –68.69±5.54 | –65.14±5.5 | –67.05±7.86 |
| Input resistance (MΩ) | 257.25±188.06 | 75.27±37.5 | 74.61±34.29 | 64.79±41.82 | 70.45±46.9 | 90.14±54.37 | 81.14±46.36 |
| Tau (ms) | 23.88±14.7 | 8.49±3.08 | 9.73±4.24 | 8.99±3.71 | 8.76±6.33 | 8.53±4.24 | 10.39±6.63 |
| Sag ratio | 0.079±0.87 | 0.075±0.058 | 0.082±0.092 | 0.086±0.076 | 0.09±0.084 | 0.092±0.073 | 0.12±0.1 |

originate from the L2/3, we measured the distance between the cell body and the L1 border (*Berg et al., 2021*). 36% of the cells recovered their soma, with a distance of 129.69±130.77 µm from the L1 border (*Figure 1E*). The data set was divided into seven age groups (*Kang et al., 2011*; *Bethlehem et al., 2022*): infant:<1 y, early childhood: 1–6 y, late childhood: 7–12 y, adolescence: 13–19 y, young adulthood: 20–39 y, middle adulthood: 40–59 y, late adulthood: ≥60 y (*Figure 1A*).

We evaluated the voltage deflections induced by negative and positive current injections and extracted subthreshold membrane features such as resting membrane potential, input resistance, time constant (tau), and sag ratio in 457 cortical pyramidal cells from 99 patients. We found that the subthreshold features from samples of infant significantly different from those from other age groups (*Figure 2A, B, C and D*; resting membrane potential: p=3.53 × 10$^{-8}$, input resistance: p=1.29 × 10$^{-16}$, tau: p=1.31 × 10$^{-15}$, sag ratio: p=5.2 × 10$^{-4}$, Kruskal-Wallis test) (*Figure 2—figure supplement 1*). The resting membrane potential was significantly more positive in the first year of life than in the other age groups. Before adulthood, a slight decrease was observed in the resting membrane potential across the groups (*Table 1*; *Figure 2A*). Input resistance, tau and sag ratio were measured on voltage deflections elicited by injecting negative (–100 pA) current steps into the cells. A significant decrease in input resistance (*Table 1*) was observed with the largest reduction after the first year of age (p=1.88 × 10$^{-6}$, Kruskal–Wallis test with post-hoc Dunn test) (*Figure 2B*). The membrane time constant also decreased significantly in the older groups compared to the infant group, after infancy we found more conserved mean values of membrane time constant into older age (*Table 1*; *Figure 2C*). The ratio of the maximal deflection and the steady-state membrane potential during a negative current step (sag ratio) is significantly higher in late adulthood than in the early stages of life (*Table 1*; *Figure 2D*). Note that the high variance of the infant group data (e.g. resting membrane potential, input resistance, tau) are due to the dynamic change over the 0–1 y period (*Figure 2—figure supplement 2*).

## Suprathreshold properties across age-groups

To initiate action potentials we injected positive current steps increased by 20 pA into the cells and recorded various types of input-output transformations. We extracted 25 features from different action potentials (AP) and firing patterns and assessed active membrane properties from recordings filtered for appropriate electrophysiological quality (see Methods). We found that the infant group differs most from the other age groups in several of the suprathreshold properties (*Figure 3*), but other trends are also apparent (rheobase current: p=8.71 × 10$^{-12}$, AP half-width: p=9.57 × 10$^{-25}$, AP up-stroke velocity: p=1.63 × 10$^{-12}$, AP amplitude: p=2.24 × 10$^{-11}$, Kruskal-Wallis test) (*Figure 3—figure supplement 1*). For example, the average rheobase current, the minimum current that can trigger an action potential, is significantly lower in the early ages of life than in data collected from adolescence stage (*Table 2*) (infant vs. adolescence p=4.15 × 10$^{-13}$, Kruskal–Wallis test with post-hoc Dunn test). Further on the age scale, we found that rheobase current was increased with age reaching a maximum value at the adolescent age and then declining to a significantly lower level (adolescent vs. late adulthood, p=5.23 × 10$^{-4}$, Kruskal–Wallis test with post-hoc Dunn test, *Figure 3A*). The AP half-width averages among a declining trend through the groups of age (*Table 2*) varied considerably across age groups forming significant differences between childhood and adulthood ages (*Figure 3B*). Action potential up-stroke velocities were significantly slower in the infant APs than in all the other ages (*Table 2*; *Figure 3C*). The amplitude of the elicited APs also showed age-dependent differences between age groups (*Table 2*);

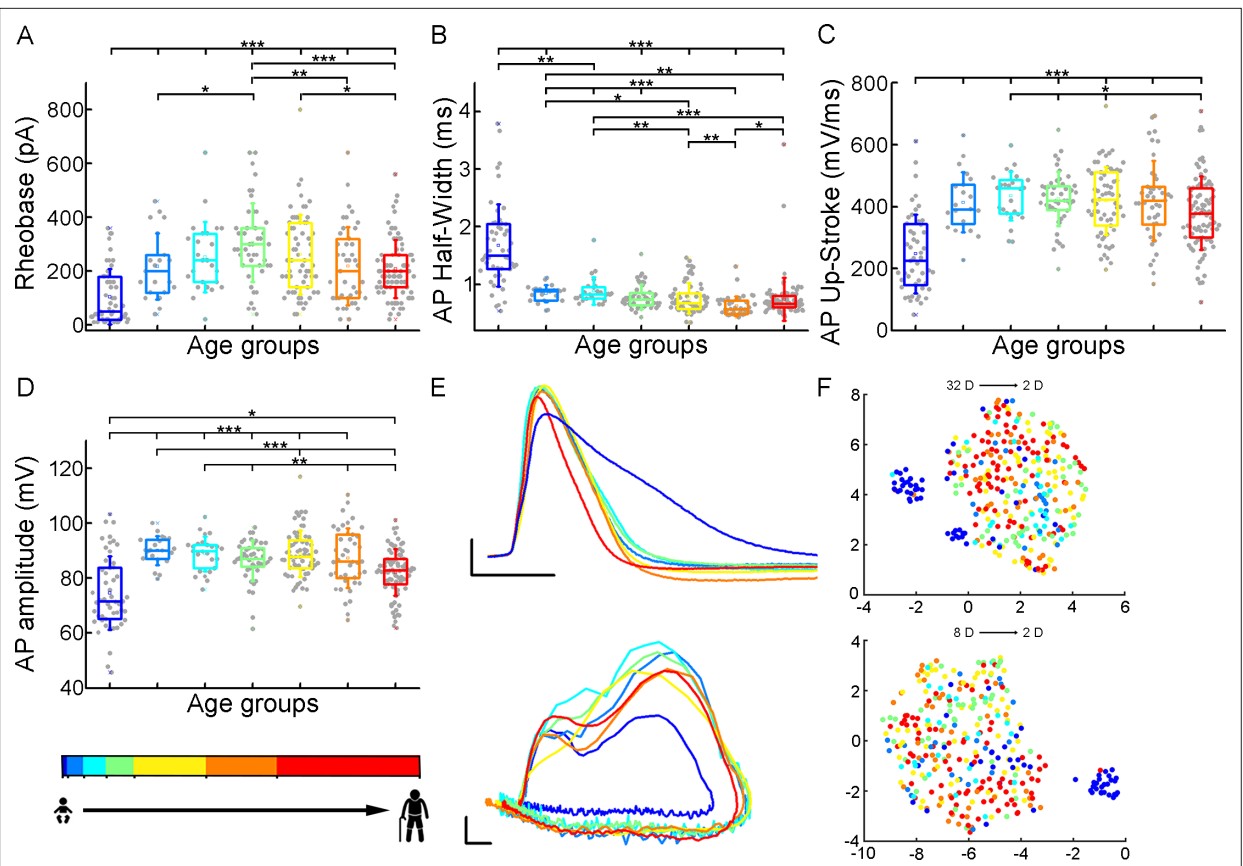

**Figure 3.** Age-related differences in the action potential kinetics. (**A–D**) Boxplots show differences in rheobase (**A**), action potential half-width (**B**), action potential up-stroke (**C**), and action potential amplitude (**D**) between the age groups. Asterisks indicate statistical significance (*p<0.05, **p<0.01, ***p<0.001). (**E**) Representative action potentials aligned to threshold potential onset (scale: x-axis: 1 ms, y-axis: 20 mV) (top) and phase plots of the representative action potentials (APs) (scale: x-axis: 10 mV, y-axis: 100 mV/ms) (bottom). (**F**) Uniform Manifold Approximation and Projection (UMAP) of 32 (*Table 4*) (top) and eight selected electrophysiological properties (resting Vm, input resistance, tau, sag ratio, rheobase, AP half-width, AP up-stroke, and AP amplitude) (bottom) with data points for 331 cortical L2/3 pyramidal cells, colored with the corresponding age groups.

The online version of this article includes the following figure supplement(s) for figure 3:

**Figure supplement 1.** Distribution of suprathreshold electrophysiological features with age.

**Figure supplement 2.** Relationship between patient metadata and electrophysiology.

**Table 2.** Action potential and firing pattern parameters.

| Suprathreshold properties | Infant N=51 | Early childhood N=21 | Late childhood N=25 | Adolescence N=45 | Young adulthood N=63 | Middle adulthood N=43 | Late Adulthood N=83 |
|---|---|---|---|---|---|---|---|
| | Mean ± SD | Mean ± SD | Mean ± SD | Mean ± SD | Mean ± SD | Mean ± SD | Mean ± SD |
| Rheobase (pA) | 104.51±103.18 | 218.1±123.27 | 252.8±131.17 | 306.22±147.21 | 262.9±148.06 | 219.07±145.52 | 207.71±107.83 |
| AP half-width (ms) | 1.68±0.71 | 0.84±0.15 | 0.88±0.24 | 0.78±0.22 | 0.76±0.27 | 0.62±0.17 | 0.74±0.38 |
| AP up-stroke (mV/ms) | 247.43±127.27 | 413.68±97.11 | 434.75±78.32 | 424.69±87.58 | 418.49±110.33 | 419.15±129.29 | 379.22±118.45 |
| AP amplitude (mV) | 74.62±13.4 | 90±5.23 | 88.82±6.35 | 86.31±7.36 | 88.81±8.52 | 87.28±10.77 | 82.02±8.49 |
| F-I slope (Hz/pA) | 0.142±0.137 | 0.152±0.066 | 0.144±0.071 | 0.139±0.141 | 0.127±0.08 | 0.176±0.113 | 0.165±0.117 |
| First AP latency (ms) | 161.26±76.81 | 121.36±76.88 | 115.29±81.27 | 136.69±109.78 | 132.42±84.68 | 140.61±143.43 | 106.39±52.31 |
| Adaptation | 0.103±0.087 | 0.061±0.062 | 0.068±0.06 | 0.105±0.1 | 0.119±0.118 | 0.114±0.153 | 0.124±0.1 |

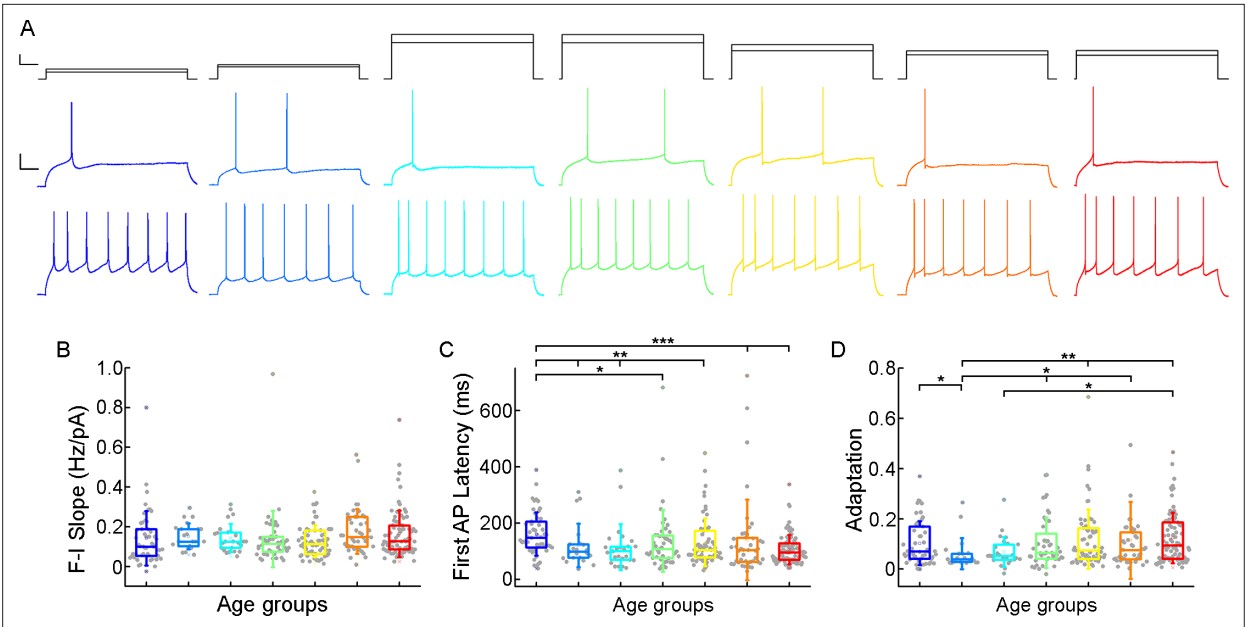

**Figure 4.** Age-dependency of the action potential (AP) firing pattern parameters. (**A**) Representative membrane potential responses to an 800 ms long rheobase (middle) (left to right: infant, early childhood, late childhood, adolescence, young adulthood, middle adulthood, late adulthood), and increased current steps (bottom) colored respectively to the age groups. Scale bar top: 1 ms, 100 pA, bottom: 1 ms, 20 mV. (**B–D**) Boxplots show changes across the age groups in f-I slope (**B**), first AP latency (**C**), and adaptation of APs (**D**). Asterisks indicate statistical significance (*p<0.05, **p<0.01, ***<0.001).

The online version of this article includes the following figure supplement(s) for figure 4:

**Figure supplement 1.** Distribution of firing pattern characteristics with age.

**Figure supplement 2.** Comparison of the subthreshold properties of the cells as a function of their distance from the layer border.

**Figure supplement 3.** Comparison of the action potential properties of the cells as a function of their distance from the layer border.

**Figure supplement 4.** Comparison of the firing pattern characteristics of the cells as a function of their distance from the layer border.

in the first year of life, the amplitude of APs were significantly lower than in other age groups. In adulthood, a significant decrease with age was observed (*Figure 3D*). Electrophysiological differences markedly separated the infant group from the older age groups shown on UMAP projection (*Figure 3F*, *Figure 3—figure supplement 2*).

Next, we investigated how somatic current inputs are transformed to action potential output by evaluating the firing patterns of cells evoked by injecting prolonged positive current steps (*Figure 4A*; *Figure 4—figure supplement 1*). The neurons (n=331) were regular spiking cells with moderate adaptation. The slope of the firing frequency versus the current curve (f-I slope) showed no significant difference across the age groups (p=0.055, Kruskal-Wallis test) (*Table 2*; *Figure 4B*). Age-related difference was observed in the latency of the first AP during rheobase current injection (first AP latency), in the first year of life the latency of the first spike is significantly higher than all the other groups of age (p=7.67 × 10$^{-4}$, Kruskal-Wallis test), which is the most prominent with the oldest age group (p=8.41 × 10$^{-6}$, Kruskal–Wallis test with post-hoc Dunn test) (*Table 2*; *Figure 4C*). The adaptation of the AP frequency response to the same current injection stimulus also showed differences (p=0.032, Kruskal-Wallis test) between the younger and the groups older than 13 y patients, we found the lowest adaptation values in early childhood (*Table 2*; *Figure 4D*).

Previous research has shown that the biophysical properties of the human pyramidal cells show depth-related correlations throughout the L2/3 (*Berg et al., 2021*; *Kalmbach et al., 2018*; *Moradi Chameh et al., 2021*). Although there are some deeper cells in our dataset, the majority comes from the upper region of the L2/3. We compared the electrophysiological characteristics according to their depth from the border of L1 and L2 to exclude the possibility that the biophysical differences we found were a result of depth dependence. We did not find any overall differences related to distance of the soma from the L1 border within the age groups with a few exceptions. For example, the values

of input resistance (p=0.02, Mann-Whitney test) and AP up-stroke velocity (p=0.04, Mann-Whitney test) differ significantly in the middle adulthood group. We found a significant difference in AP amplitude (p=0.02, Mann-Whitney test) and adaptation (p=0.009, Mann-Whitney test) in the adolescence age group (*Figure 4—figure supplement 2*, *Figure 4—figure supplement 3* and *Figure 4—figure supplement 4*).

## Morphological features of layer 2/3 pyramidal cells in different stages of life

To investigate possible morphological differences between the age groups, we filled the pyramidal cells with biocytin during recordings. Only neurons with no signs of deterioration and with complete apical dendrites and no signs of truncated dendritic branches or tufts were considered for morphological analysis. 63 pyramidal cells (*Figure 5—figure supplement 1*) were reconstructed in 3D at ages 0–73 y (infant n=7, early childhood n=8, late childhood n=11, adolescence n=11, young adulthood n=9, middle adulthood n=9, late adulthood n=8) (*Figure 5—figure supplements 2 and 3*). *Figure 5A* shows examples of the reconstructed pyramidal cells. We did not detect significant change in total dendritic length (p=0.37, Kruskal-Wallis test) (*Table 3*; *Figure 5B*), apical dendritic length (p=0.6, Kruskal-Wallis test) (*Table 3*; *Figure 5C*), or basal dendritic length (p=0.28, Kruskal-Wallis test) (*Table 3*; *Figure 5D*) at different ages. To investigate dendritic complexity we measured the total number of dendritic branching and found no significant developmental changes (p=0.18, Kruskal-Wallis test) (*Table 3*; *Figure 5E*). We also did not observe significant differences in the size of dendritic branching when we measured the maximum horizontal (p=0.64, Kruskal-Wallis test) (*Table 3*; *Figure 5F*) and vertical (p=0.51, Kruskal-Wallis test) (*Table 3*; *Figure 5G*) extent of the reconstructed cells. We found significant differences in the average length of the cut terminal segments of apical (p=0.033, Kruskal-Wallis test) (*Table 3*; *Figure 5H*) but not of the basal dendrites (p=0.85, Kruskal-Wallis test) (*Table 3*; *Figure 5I*) of the cells across the age groups .

To analyze the distribution of dendritic spines we identified and labeled each spine on n=6 fully 3D-reconstructed cells (*Figure 6*, *Supplementary file 2*). We compared the spine density of selected pyramidal cells of two age groups: infant (83 d old, n=3 of one patient, parietal lobe) vs. late adulthood (64.3±2.08 y old, n=3 of 3 patients, frontal, temporal and parietal lobes). The investigated cells are located in L2 (infant: 144.43±45.26 µm, late adulthood: 161.22±66.22 µm). We found that the total spine density was higher (p=7.57 × 10$^{-40}$, Mann-Whitney test) and also the spine density of both apical (p=2.02 × 10$^{-31}$, Mann-Whitney test) and basal (p=3.8 × 10$^{-12}$, Mann-Whitney test) dendrites was higher in the infant than in the late adult group (*Figure 6B*). To evaluate the age-dependence of spine morphology, we classified the spines into commonly used phenotypes based on their morphological characteristics (*Li et al., 2023*), specifically distinguishing between mushroom-shaped, thin, filopodial, branched, and stubby spines (*Figure 6C–G*). Spines with large spine heads were classified as mushroom-shaped, those with small heads as thin, and long protrusions were distinguished as filopodial. Those that did not have peduncles were classified as stubby, spines with two heads emerging from the same spot were called branched spines (*Figure 6C–G*; *Luebke et al., 2015*). Only fully visible spines were included in the classification analysis. The composition of spine types varies between the two age groups, mushrooms are present at a higher percentage on apical branches in late adulthood cells (p=4.4 × 10$^{-9}$, Mann-Whitney test), and on the basal processes (p=9.04 × 10$^{-8}$, Mann-Whitney test) (*Figure 6C and H*). In contrast, thin spines and filopodia are present in significantly higher numbers on the apical (*Figure 6D and E*) (thin spines: p=7.34 × 10$^{-14}$; filopodia: p=1.11 × 10$^{-39}$, Mann-Whitney test) and basal (thin: p=2.46 × 10$^{-8}$; filopodia: p=2.14 × 10$^{-12}$, Mann-Whitney test) (*Figure 6I–J*) dendritic branches of the infant. Both apical and basal infant branches had a significantly higher percentage of branched spines (apical: p=1.64 × 10$^{-11}$; basal: p=8.9 × 10$^{-5}$, Mann-Whitney test) (*Figure 6F and K*), which were present in modest numbers on both. Stubby spines were also more prevalent on the elderly pyramidal cells, either on the apical (p=7.19 × 10$^{-5}$, Mann-Whitney test) or basal (p=6.97 × 10$^{-9}$, Mann-Whitney test) processes (*Figure 6G and L*). Comparing the spine density of the six individual cells, we found differences across the two age groups similar to those previously mentioned, alongside slight variation within age groups. (*Supplementary file 2*, Kruskal-Wallis test with post-hoc Dunn test).

Cortical tissue was dissected during neurosurgical procedures for various pathologies, but not to the same extent in the different groups. Specimen collection in adult/elderly patients was mostly for

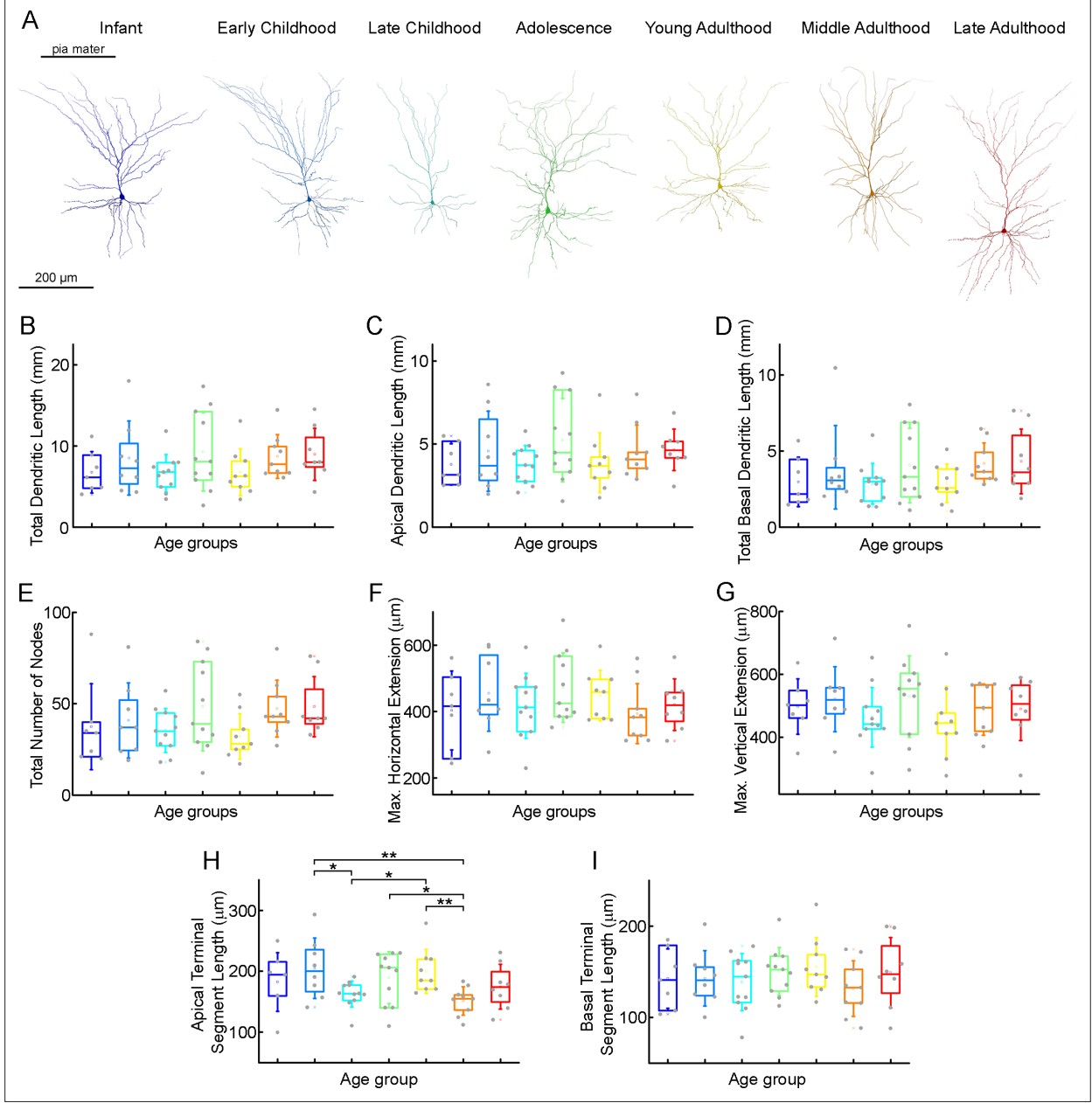

**Figure 5.** Morphological features of layer 2/3 pyramidal cells in different stages of life. (**A**) Representative reconstructions of L2/3 pyramidal cells (from left to right) from infant (n=7), early childhood (n=8), late childhood (n=11), adolescence (n=11), young adulthood (n=9), middle adulthood (n=9), and late adulthood (n=8) patients. (**B–I**) Boxplots show summarized data from all the reconstructed cells (*Figure 4—figure supplement 2*) of total dendritic length (**B**), apical dendritic length (**C**), total basal dendritic length (**D**), the total number of nodes on the apical and basal dendrites (**E**), the maximal horizontal (**F**), and the maximal vertical (**G**) extension of dendrites, the average length of the apical (**H**), and basal (**I**) terminal dendritic segments. Asterisks indicate statistical significance (*p<0.05, **p<0.01, Kruskal-Wallis test with post-hoc Dunn test).

The online version of this article includes the following figure supplement(s) for figure 5:

**Figure supplement 1.** Human cortical L2/3 pyramidal cell dendritic reconstructions.

**Figure supplement 2.** Morphological comparison of the examined infant cells.

**Figure supplement 3.** Distribution of morphological features with age.

tumor resection, whereas in children hydrocephalus was the most common reason for brain surgery (*Figure 1—figure supplement 1*). To determine the difference in medical condition could contribute to age-related differences in cellular features, we compared the extracted electrophysiological and morphological features based on the medical condition in the different age groups. It should be

**Table 3.** Morphological characteristics across the age groups.

| Morphological properties | Infant N=7 Mean ± SD | Early childhood N=8 Mean ± SD | Late childhood N=11 Mean ± SD | Adolescence N=11 Mean ± SD | Young adulthood N=9 Mean ± SD | Middle adulthood N=9 Mean ± SD | Late adulthood N=8 Mean ± SD |
|---|---|---|---|---|---|---|---|
| Total dendritic length (mm) | 6.74±2.56 | 8.5±4.56 | 6.62±2.29 | 9.27±4.82 | 6.76±2.9 | 8.7±2.7 | 8.96±3.22 |
| Apical dendritic length (mm) | 3.77±1.24 | 4.57±2.42 | 3.78±1.14 | 5.24±2.51 | 3.88±1.8 | 4.52±1.63 | 4.64±1.25 |
| Basal dendritic length (mm) | 2.97±1.63 | 3.93±2.74 | 2.84±1.33 | 4.04±2.44 | 2.88±1.27 | 4.18±1.35 | 4.31±2.13 |
| Total number of nodes | 37.42±23.58 | 40.88±20.54 | 35.36±12.07 | 48.36±24.23 | 32.11±12.38 | 47.33±15.57 | 48.38±16.47 |
| Max. horizontal extension (μm) | 403.33±118.55 | 455.24±114.42 | 417.2±98.1 | 472.2±104.51 | 448.3±75.94 | 393.7±90.29 | 420.93±77.49 |
| Max. vertical extension (μm) | 498.12±87.85 | 521.21±102.88 | 463.83±94.63 | 529.58±129.28 | 447.41±114.17 | 488.02±81.02 | 490.45±100.53 |
| Apical terminal length (μm) | 182.42±48.17 | 204.83±49.62 | 162.47±21.5 | 189.59±42.54 | 199.68±36.2 | 151.23±23.3 | 174.62±36.76 |
| Basal terminal length (μm) | 142.54±32.43 | 142.69±30.31 | 138.7±31.22 | 150.32±26.09 | 155.29±32.15 | 131.6±30.44 | 148.96±38.42 |

noted that all circumstances of surgical steps, tissue dissection, transport (time, media, temperature etc.) and cutting procedure remained the same in the different conditions. When comparing passive electrophysiological properties, we found no significant differences between the different medical conditions in the age groups from infancy to middle adulthood, only time constants were found to be significantly lower in the hydrocephalus patients than in the tumor patients in the young adulthood (p=0.048, Mann-Whitney test) and late adulthood groups (p=0.01, Mann-Whitney test) (*Figure 6—figure supplement 2*). Comparing the action potential kinetics and firing pattern-related parameters between the different pathology groups we found no overall differences with some sporadic exceptions. For example, the parameters of AP half-width in young adulthood (p=0.01, Mann-Whitney test) and the values of rheobase in early childhood (p=0.04, Mann-Whitney test) differed significantly between the different pathologies (*Figure 6—figure supplement 2*). Regarding the firing pattern-related properties we found significant changes in the infant F-I slope (p=0.02, Mann-Whitney test), first AP latency (p=0.006, two-sample t-test) and adaptation (p=0.002, Mann-Whitney test) parameters. Also in the late adulthood group the F-I slope (p=0.04, Mann-Whitney test), the first AP latency (p=0.02, Mann-Whitney test) and the adaptation (p=0.047, Mann-Whitney test) were significantly different between the hydrocephalus and tumor patients (*Figure 6—figure supplement 4*). Further comparison of morphological features revealed no statistical difference between tumor and hydrocephalus groups (*Figure 6—figure supplement 5*).

To ascertain whether age-related variations in cell properties may be influenced by the gender of the patient, we compared the examined characteristics of the cells by their sex (*Figure 6—figure supplements 6 and 7* and 8). Comparing the passive membrane properties of the cells in different age groups we found higher resting membrane potential values in adolescent females than in males (p=0.04, Two-sample t-test) and input resistance was lower in female patients in the late childhood group (p=0.03, Mann-Whitney test) (*Figure 6—figure supplement 6*). We found differences when comparing the AP characteristics in AP half-with cells from male patient showed higher values in early childhood (p=0.036) and adolescence (p=0.02). AP up-stoke velocity was higher in young adulthood females than in males (p=0.34). AP amplitude was significantly higher in male patients in early childhood (p=0.003) and late adulthood (p=0.035) age groups (*Figure 6—figure supplement 7*). Under the comparison of the firing pattern features cells from the infancy showed significant differences in F-I slope (p=0.2, Mann-Whitney test), first AP latency (p=0.0096, Two-sample t-test), and adaptation (p=0.002, Mann-Whitney test). The latency of the first AP was higher in cells from male patient in the young (p=0.03, Mann-Whitney test) and middle (p=0.018, Mann-Whitney test) adulthood cells. The

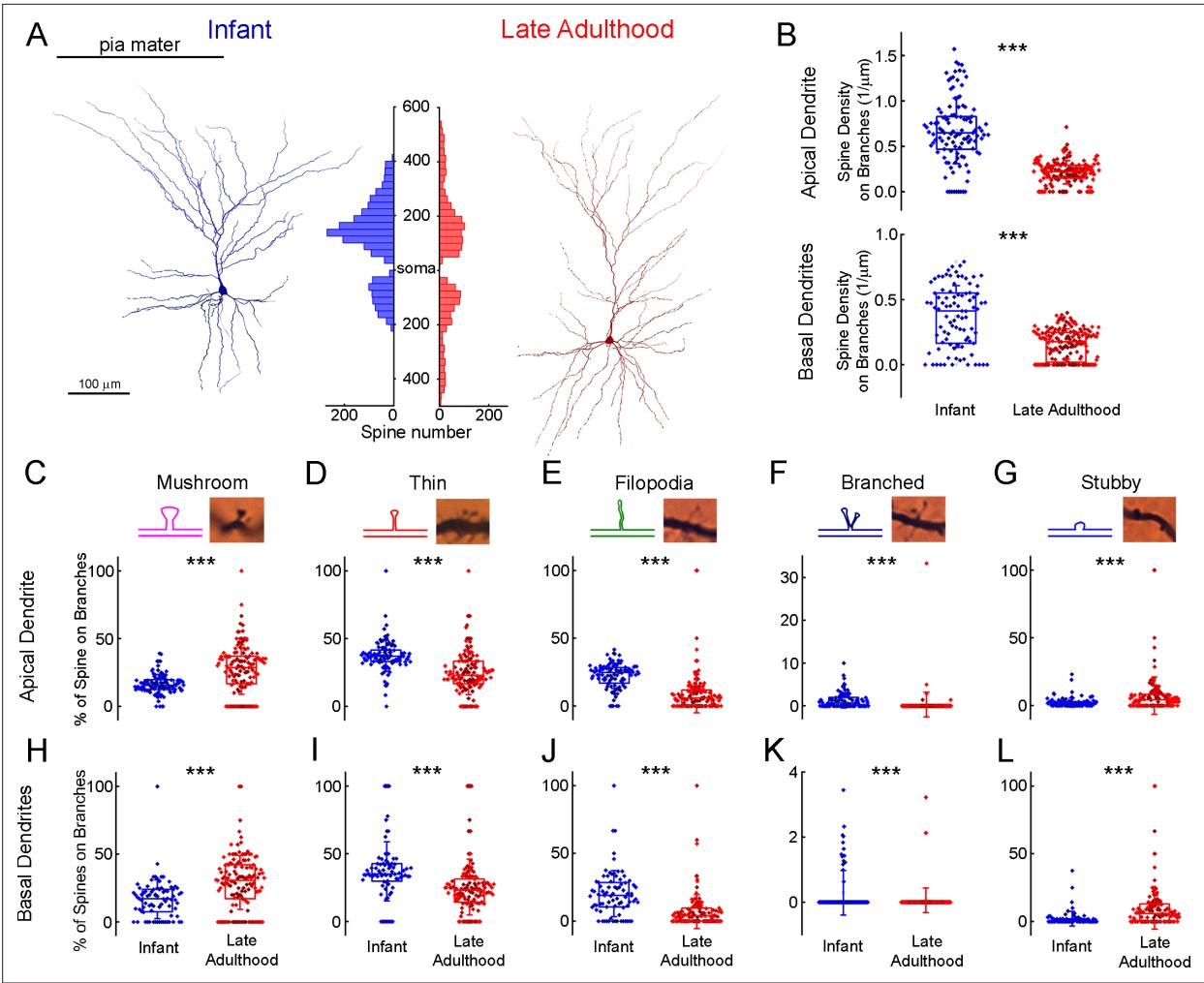

**Figure 6.** Comparison of dendritic spine densities in pyramidal cells from infant and late adulthood samples. (**A**) Anatomical 3D reconstruction of human L2/3 pyramidal cells from the infant (left), and the late adulthood (right) age groups. The histogram (in the middle) demonstrates the distribution of dendritic spines on the two representative cells according to their distance from the soma (μm) on the apical and basal dendrites. (**B**) Boxplots of the average spine densities on the apical (top), and basal (bottom) dendritic branches from the n=3 infant (blue) and n=3 late adulthood (red) L2/3 pyramidal cells. The symbols are color-coded by the 6 individual cells.(**C–G**) The plots show the distribution of mushroom (**C**), thin (**D**), filopodium (**E**), branched (**F**), and stubby (**G**) dendritic spine types on the apical dendrites of the reconstructed infant (n=3, blue) and late adult (n=3, red) pyramidal cells. Top, schematic illustration and representative images of the examined dendritic spine types. Center, age-dependent distribution of spine types. Asterisks indicate significance (*p<0.05, **p<0.01, ***p<0.001). (**H–L**) Same as C-G but on basal dendrites.

The online version of this article includes the following figure supplement(s) for figure 6:

**Figure supplement 1.** Comparison of dendritic spine densities in pyramidal cells from all samples.

**Figure supplement 2.** Comparison of subthreshold electrophysiological properties of the examined cells from patients with tumor removal or ventriculoperitoneal (VP) shunt surgical procedures.

**Figure supplement 3.** Comparison of action potential properties of the examined cells from patients with tumor removal or ventriculoperitoneal (VP) shunt surgical procedures.

**Figure supplement 4.** Comparison of firing pattern characteristics of the examined cells from patients with tumor removal or ventriculoperitoneal (VP) shunt surgical procedures.

**Figure supplement 5.** Morphological comparison of the examined cells from patients with tumor removal or ventriculoperitoneal (VP) shunt surgical procedures.

**Figure supplement 6.** Comparison of subthreshold electrophysiological properties of the examined cells from female and male patients.

**Figure supplement 7.** Comparison of action potential properties of the examined cells from female and male patients.

**Figure supplement 8.** Comparison of firing pattern characteristics of the examined cells from female and male patients.

**Figure supplement 9.** Morphological comparison of the examined cells from female and male patients.

adaptation of the APs in males was higher during adolescence (p=0.027, Mann-Whitney test) but lower in middle adulthood (p=0.034, Mann-Whitney test) (*Figure 6—figure supplement 8*). Regarding the morphological property of the reconstructed cell we found differences in the young adulthood age group between cells from male and female patients in total (p=0.03, Mann-Whitney test) and apical (p=0.03, Mann-Whitney test) dendritic length and the total number of bifurcations (p=0.03, Mann-Whitney test), in the middle adulthood group we found significant difference in the average apical terminal length across males females(p=0.03, Mann-Whitney test) (*Figure 6—figure supplement 9*).

## Discussion

In this study, using the whole-cell patch-clamp technique and 3D reconstructions, we have studied the differences of human cortical L2/3 pyramidal cells at various stages of life. We found that during the lifespan the most significant changes take place early in life during the first year in most of the biophysiological characteristics of human pyramidal cells (see *Figure 2—figure supplement 2*). We recorded from n=457 human cortical excitatory pyramidal cells from the supragranular layer from birth to 85 y. Most differences in sub- and suprathreshold features were found in the youngest age groups. There were particular differences in resting membrane potential, input resistance, time constant, rheobasic current, AP halfwidth, and AP upstroke velocity. Other age groups that differed the most from other age groups were the oldest (60–85 y) with modest differences in sag ratios. In our morphological analysis, we found no significant changes in the overall apical and basal dendritic dimensions across different ages. When evaluating the number of spines in two anatomical reconstructions (from the infant and late adulthood groups) we found a higher overall spine density in the infant than in the elder sample consisting of mainly branched and thin spines and filopodia. In contrast, stubby and mushroom-shaped spines were more prevalent in the older pyramidal cell.

The nervous system experiences a multitude of plastic changes throughout an individual's lifetime. These changes extend from the early stages of development and continue through the gradual degenerative processes that come with old age. Although precise intrinsic cellular modifications linked to aging remain somewhat elusive, there is an observable morphological transformation during early development. As the nervous system matures, dendritic branching (*Kasper et al., 1994*; *McAllister, 2000*) and cell body size increases (*Zhang, 2004*). Also, learning and experience increase the extent of dendritic branching (*Holloway, 1966*; *Greenough and Volkmar, 1973*) and dendritic spines (*Globus et al., 1973*). An extensive body of research has shown that synaptic plasticity induces formation of new dendritic spines and the enlargement of existing spines (*Kastellakis et al., 2023*). These modifications may affect the input-output relation of a neuron as model studies suggest that dendritic geometry influences electrical and firing properties (*Mainen and Sejnowski, 1996*; *van Elburg and van Ooyen, 2010*; *Eyal et al., 2014*; *Vetter et al., 2001*). Furthermore, in parallel to the changes in morphological complexity, there is ample experimental evidence that the maturation of ion channels and the change in increasing channel density in the neuronal membrane also play a role in influencing the electrical properties of neurons (*Huguenard et al., 1988*; *Connors, 1994*). These findings point to a close causal relationship between structure and electrophysiological properties at cellular level.

### Lifespan changes in electrophysiological properties

The intrinsic electrophysiological properties of a neuron are determined by several factors: the resistivity of the cytoplasm and membrane, the membrane capacitance, and the shape of the soma and dendrites. The basic passive parameters we recorded, such as the resting membrane potential, the input resistance, and the membrane time constant show a dramatical change in the first year. This change is consistent with data found in previous studies on neocortical pyramidal cells from rodent models (*Kasper et al., 1994*; *Kroon et al., 2019*; *Picken Bahrey and Moody, 2003*; *Zhang, 2004*; *Zhu, 2000*; *McCormick and Prince, 1987*) and on xenotransplanted human cortical neurons (*Linaro et al., 2019*). We found a progressive shift of the resting membrane potential from –60 to –68 mV in the first part of the lifespan (<40 y), which could be due to the age-dependent change in ion channel composition (*Picken Bahrey and Moody, 2003*). The input resistance and time constant showed a sudden and sharp decrease between the first and second age groups, indicating the most dramatic changes in the infant's brain followed by a generalized and slower decrease. As pyramidal cells mature, the size and volume of the cell body increase (*Petanjek et al., 2019*; *Zhu, 2000*) which

correlates with the input resistance, meaning that a small membrane area has a higher input resistance (*Luhmann et al., 2000*). In addition to the increase in cell surface area, the increase in the expression of certain potassium leak channels, e.g., from the two-pore domain potassium channel (KCNK) family, can also influence the input resistance during cell maturation (*Aller and Wisden, 2008*; *Goldstein et al., 2001*). Similar reasons can lead to a decrease in the membrane time constant. The membrane time constant is the product of the specific input resistance and the specific membrane capacitance. Assuming constant membrane capacitance (~1 pF/cm$^2$, *Oláh et al., 2024*, *Beaulieu-Laroche et al., 2018*), the decrease is likely also due to a change in specific membrane resistance. The result of developmental changes in membrane passive parameters during early development improves spike timing precision and temporal responsiveness in the cortical microcircuit (*Oswald and Reyes, 2011*; *Doischer et al., 2008*). However, in our human samples at later ages (>40 y) that we associate with the aging brain, the previous trends of changes in intrinsic characteristics are reversed. Input resistance and time constant increase, which may be associated with the decline in KCNK expression in the aging brain (*Erraji-Benchekroun et al., 2005*). These membrane parameters are also generally higher in studies in rhesus monkeys (*Moore et al., 2023*), except for the resting membrane potential, which is not different from that of young cortical L2/3 (*Chang et al., 2005*) or L5 pyramidal cells (*Luebke and Chang, 2007*) in the brain. However, in our samples, we found a positive shift in the resting potential between the middle- and old-age groups by a few millivolts. It should be noted that this discrepancy may be due to the fact that the age groups in our work are not the same as the works mentioned.

Changes in resting membrane potential and input resistance will add up to produce a rising and then falling curve of the averaged rheobasic current, with the value three times higher in the adolescent group than in the youngest group and 1.5 times higher than in late adulthood. Our results on increased rheobase during early life are consistent with previous studies in rodent cortical L3 and 5 pyramidal cells (*Kroon et al., 2019*; *Perez-García et al., 2021*; *Popescu et al., 2021*) and with findings regarding aging in rhesus monkey cortical neurons from rhesus monkeys (*Moore et al., 2023*).

When we assessed the voltage sag ratio with age we found a consistent increase that peaked in late adulthood. The voltage sag that occurs during membrane hyperpolarization is the result of activation of the hyperpolarization-activated cyclic nucleotide-gated (HCN) current, also known as the h-current. Similar to our findings the age-dependent increase in sag deflection has been observed in human cortical pyramidal cells (*Guet-McCreight et al., 2023*; *Kalmbach et al., 2018*) which is consistent with studies by *Wang et al., 2007*; *Wang et al., 2011* that have linked inhibited HCN activation to working memory deficits in the elderly.

The electrical output of neurons is driven by the composition of neuronal molecules and ion channels whose expressions and distributions change most dynamically in two periods, early development, and the aging brain. This dynamic essentially produced an inverted U-shaped curve in the distribution of the active electrical properties over the lifespan in our dataset, an increase in the first year, a relatively stable middle phase, and then a return in older age groups. The origin of these changes is mainly due to changes in the distribution of sodium and potassium channels. During development, there is evidence that the number of voltage-gated Na$^+$ channels increases in rodents (*Huguenard et al., 1988*; *Picken Bahrey and Moody, 2003*) and in humans (*Erraji-Benchekroun et al., 2005*), which may accelerate the kinetics of the action potential (up-stroke) and, similar to our results, increase the amplitude of the action potential (*Perez-García et al., 2021*; *Kroon et al., 2019*; *Zhang, 2004*; *McCormick and Prince, 1987*; *Etherington and Williams, 2011*). During aging, however, regulation is disrupted by the change in channel expression, which can lead to altered conduction rate of ionic currents. In brain senescence, the amplitude of the action potential and a slowing of the action potential *Luebke and Chang, 2007* have been shown to be similar to our data, which may be caused by the decreased expression of voltage-gated Na$^+$ channels (*Erraji-Benchekroun et al., 2005*).

We examined the firing patterns of pyramidal cells at different ages by comparing the F-I slope, and we found that the patterns are fairly conserved across ages. In rodent developmental studies, a decrease in F-I slope was found only in the first 2 wk after birth (*Zhang, 2004*; *McCormick and Prince, 1987*), which time window is not represented in our data. The latency of the first spike is influenced by the time-constant and specific ion channels, like the transient (or A-type) K$^+$ current and the transient Ca$^{2+}$ current (or T-type) (*Molineux et al., 2005*). The same ion channels are involved in burst firing at the onset of stimulation. Note that in our sample only ~10% of pyramidal cells showed burst firing at the beginning of stimulation, but it is known that burst firing is more frequent

from layer 3 onwards at depth (*Berg et al., 2021*). The presence of A-current has been found in immature pyramidal neurons from sensorimotor areas in rodent studies (*McCormick and Prince, 1987*), and the density of T-type currents also remains unchanged during the development of the visual cortex (*Horibe et al., 2014*). It has been shown that the expression of the T-type calcium channel family (CACNG3) decreases during aging (*Erraji-Benchekroun et al., 2005*). However, it must be mentioned that the differences in the expression of all subtypes of the Kv4 (A-type) and Ca$_v$3.x (T-type) channel families during brain development and aging have not been systematically described.

## Properties of dendritic trees and spine distribution with age

Transcriptomic analyses of the human neocortex show relatively stable expression patterns of genes for dendritic and synaptic development after about 1 y of age (*Kang et al., 2011*). In early post-mortem studies, human pyramidal neurons were studied for morphological changes as a function of age. In these studies, using Golgi staining methods, it was found that the dendritic trees of infant layer 2/3 pyramidal cells are well developed (*Mrzljak et al., 1990*). In the human prefrontal cortex after birth, the dendritic tree of layer 3 pyramidal neurons reaches its structurally mature form after about 3 m (*Petanjek et al., 2008*). Thereafter, it is assumed that a smaller increase in dendritic length reaches its final form between about 7.5–12 m (*Koenderink and Uylings, 1995*; *Petanjek et al., 2008*). In our study, we observed stable patterns of apical and basal dendrite lengths and dendritic tree complexity across the lifespan. Even between the early postnatal groups (infancy vs. early childhood), no significant difference in dendritic tree size was observed. This can be explained by the influence of various factors, such as the large difference between individual subjects or the subtype-specific dendritic morphology of pyramidal cells (*Berg et al., 2021*) or different maturation curves of neurons from different brain regions (*Elston and Fujita, 2014*) or cortical layers (*Petanjek et al., 2008*).

Neurons exhibit a phase of synaptogenesis overproducing synapses lasting months or years in the human cortex (*Huttenlocher, 1979*; *Huttenlocher et al., 1982*; *Mrzljak et al., 1990*; *Petanjek et al., 2011*), followed by dendritic spine/synapse pruning that is reported to last more than a decade in the cortex (*Petanjek et al., 2011*; *Jacobs et al., 1997*; *Mavroudis et al., 2015*; *Benavides-Piccione et al., 2013*; *Coskren et al., 2015*). Our results from comparing two groups: infant and late adult pyramidal cells also show that the overall number of dendritic spines decreases with increasing age. When evaluating the distribution of spine shapes in young and old pyramidal cells, we found spines with larger heads (mushroom shape) in a greater number in the dendrite of old pyramidal cells, which are considered as mature synapses. The size of the head is an indicator of the size of the postsynaptic density, the number of glutamate receptors, and the strength of the synapse (*Bourne and Harris, 2008*; *Matsuzaki et al., 2001*; *Nusser et al., 1998*). In the developing nervous system, however, the filopodia, thin dendritic protrusions without a head, are the most characteristic type of spines. The density of filopodial spines was higher on the dendrites of infant pyramidal cells. In the older cells, we still found filopodia, which are the silent precursor of active synapses (*Vardalaki et al., 2022*). The change in the ratio of subtypes from young to old age is well established and is associated with the basis for changes in cognitive function during aging (*Luebke et al., 2015*; *Dumitriu et al., 2010*; *Jacobs et al., 1997*).

The complexity of human brain activity and cognitive abilities increases with development and decreases with aging, resulting in an inverted U-shaped curve across the lifespan (*Craik and Bialy-stok, 2006*). Cognitive functions depends on many age-dependent factors, such as the density and specificity of synaptic connections formed by synaptic pruning and plasticity, or the degree of myelin-ation and white matter maturation and there is also an age-related modulation of neurotransmitters, hormones, ion transporters, and receptors (*Luna et al., 2015*). Here, we have shown that pyramidal cells become less excitable and temporarily more precise during development by changing their intrinsic functional properties and with aging these changes occur somewhat in opposite directions making the intrinsic parameters change symmetrically. In addition, some of the changes are asymmetrical either occur with development or with aging, such as the resting membrane potential which mostly changes in young ages or the ratio of sag, which is shifted most in old age. These changes in the intrinsic properties of cells during the first and last stages of life also contribute to the input-output functions of a neuron and ultimately to the age-related development of cognitive abilities.

## Materials and methods

### Slice preparation

Experiments were performed according to the Declaration of Helsinki with the approval of the University of Szeged Ethical Committee and Regional Human Investigation Review Board (ref. 75/2014). Prior to surgery, the patients provided written consent for all tissue material. We used human cortical tissue adjacent to the pathological lesion that had to be surgically removed from patients (n=64 female n=45 male) as part of the treatment for tumors, hydrocephalus, apoplexy, cysts, and arteriovenous malformation. Anesthesia was induced with intravenous midazolam and fentanyl (0.03 mg/kg, 1–2 μg/kg, respectively). A bolus dose of propofol (1–2 mg/kg) was administered intravenously. The patients received 0.5 mg/kg rocuronium to facilitate endotracheal intubation. The trachea was intubated, and the patient was ventilated with a mixture of $O_2$-$N_2O$ at a ratio of 1:2. Anesthesia was maintained with sevoflurane at a care volume of 1.2–1.5. During the surgical procedure tissue blocks were removed from parietal (n=22), temporal (n=35), frontal (n=42), and occipital (n=10) regions, the resected tissue blocks were immediately immersed in ice-cold solution. Slices were cut perpendicular to the pia mater at a thickness of 320 μm with a vibrating blade microtome (Microm HM 650 V) in ice-cold solution (in mM) 75 sucrose, 84 NaCl, 2.5 KCl, 1 $NaH_2PO_4$, 25 $NaHCO_3$, 0.5 $CaCl_2$, 4 $MgSO_4$, 25 D(+)-glucose, saturated with 95% O2 and 5% CO2. The slices were incubated in the same solution for 30 min at 36 °C following that the solution was changed to (in mM) 130 NaCl, 3.5 KCl, 1 $NaH_2PO_4$, 24 $NaHCO_3$, 1 $CaCl_2$, 3 $MgSO_4$, 10 D(+)-glucose, saturated with 95% $O_2$ and 5% $CO_2$, the slices were kept in it until use.

### In vitro electrophysiological recordings

Somatic whole-cell current-clamp recordings were obtained at ~36 °C in solution containing (in mM) 130 NaCl, 3.5 KCl, 1 $NaH_2PO_4$, 24 $NaHCO_3$, 3 $CaCl_2$, 1.5 $MgSO_4$, 10 D(+)-glucose, from layer 2/3 pyramidal cells visualized by infrared differential interference contrast (DIC) video microscopy equipped with micromanipulators (Luigs and Neumann, 652 Ratingen, Germany) and HEKA EPC 9&10 patch clamp amplifier (HEKA Elektronik GmbH, Lambrecht, Germany). Micropipettes (3–5 MΩ) were filled with intracellular solution containing (in mM) 126 potassium-gluconate, 4 KCl, 4 ATP-Mg, 0.3 GTP-$Na_2$, 10 HEPES, 10 phosphocreatine, and 8 biocytin (pH 7.20; 300 mOsm). After whole-cell configuration was obtained stepwise currents were injected to measure the evoked sub- and suprathreshold membrane potential properties. For the analysis of the electrophysiological recordings n=457 recordings with a series resistance (Rs) of 24.93±11.18 MΩ (max: 63.77 MΩ) were used. For the analysis of fast parameters related to the action potential (AP half-width, AP upstroke velocity, AP amplitude, and rheobase), higher quality requirements were set and cells with Rs >30 MΩ were excluded. This reduced the data set to n=331 cells with Rs 19.42±6.2 MΩ.

### Data analysis

Electrophysiological features (*Table 4*) were measured from voltage responses elicited by 800 ms long current steps increasing by 20 pA from –100 pA. We analyzed the electrophysiological data with Fitmaster software (HEKA Elektronik GmbH, Lambrecht), and custom MATLAB (The Math Works, Inc) scripts.

Analysis of morphological features was made by NeuroExplorer software (MBF Bioscience, Williston, VT, USA) and Origin 9 (OriginLab, Northampton, MA).

### Statistics

We used custom-written R scripts (R 4.1.2) and the gamm function from the mgcv R package (mgcv 1.8.38). Data presented as the mean ± s.d. Normality was tested with the Lilliefors test, for statistical analysis, ANOVA with posthoc Bonferroni test, Kurskal-Wallis with posthoc Dunn test, for pairwise comparison two-sample t-test or Mann-Whitney test was used. Differences were accepted as significant if $p < 0.05$. The data are shown on boxplots, boxes indicate 25th, 50th (median), and 75th percentiles, rectangle represents the mean value, and whiskers indicate s.d.

### Histology and reconstruction

Slices were fixed in a fixative of 4% paraformaldehyde, 15% picric acid, and 1.25% glutaraldehyde in 0.1 M phosphate (PB, pH = 7.4) for at least 12 hr after electrophysiological recording. After multiple

**Table 4.** Examined electrophysiological properties.

| | |
|---|---|
| Resting Vm | The membrane potential of the neuron, measured directly after attaining the whole-cell configuration with no current (if a holding current was used during the recording we compensated the resting membrane potential with the injected current). |
| Input resistance | To calculate input resistance the mean of all hyperpolarizing current produced voltage steps were used. |
| Tau | To calculate time constant the mean of all hyperpolarizing current produced voltage steps were used, measured between 0–63%. |
| Sag ratio | The ratio of the maximal deflection and the steady-state membrane potential during a –100 pA current step. |
| Rheobase | The minimal current step that elicited the first spike. |
| AP half-width | The width of the AP at half amplitude. |
| AP up-stroke | The mean of all the maximum values of dV/dt between the action potential onset and the action potential peak from each elicited APs of the cell. |
| AP amplitude | Average amplitude of all APs, from threshold to peak. |
| F-I slope | The slope of the line fitted to the data of the AP firing frequency versus stimulus intensity. |
| First AP latency | The duration from the start of the stimulus until the first AP under the rheobasic current step. |
| Adaptation | The average adaptation of the interspike interval between consecutive APs. |
| Rebound | The difference between the steady-state membrane potential and the maximum deflection after a hyperpolarizing current step. |
| Rebound-Sag ratio | The ratio of the rebound and sag amplitudes. |
| Avg. AP number | Average number of elicited APs per sweep. |
| AP threshold | Mean of all the voltage values at AP threshold. |
| Ap rise time | Mean time between the threshold and the peak of all APs. |
| AHP amplitude | Average amplitude of all afterhyperpolarization. |
| AHP length | Average duration of 0 (AHP minimum) to 90% of all the AHP. |
| Voltage at max.dV/dt | Average voltage value at the maximum of the AP dv/dt over all APs |
| Velocity at min. dV/dt | Average velocity value at the minimum of the AP dv/dt from all APs |
| Voltage at min. dV/dt | Average voltage value at the minimum of the AP dv/dt from all APs |
| AP peak | Average of AP maximum voltages |
| ISI mean | Average interspike interval from all sweeps containing at least three APs |
| AP amplitude accommodation | The difference of the first and last AP amplitude in a sweep. |
| AP half-width accommodation | The difference of the first and last AP half-width in a sweep. |
| AP threshold accommodation | The difference of the first and last AP threshold in a sweep. |
| Average ISI | The mean value of interspike intervals in a sweep. |
| AP amplitude adaptation | The average adaptation of the AP amplitude between consecutive APs. |
| AP half-width adaptation | The average adaptation of the AP half-width between consecutive APs. |
| AP threshold adaptation | The average adaptation of the AP threshold between consecutive APs. |
| AHP area | Integral of the AHP from the minimum value to 90%, using trapezoidal method. |
| ADP amplitude | The amplitude of the afterdepolarization, the average difference between the minimum value of the AHP and the threshold of the next AP. |

washes in 0.1 M PB, slices were cryoprotected in 10% then 20% sucrose solution in 0.1 M PB. The slices were frozen in liquid nitrogen and then thawed in PB. Slices were embedded in 10% gelatin and further sectioned into 70 μm thick sections. Sections were incubated in a solution containing conjugated avidin-biotin horseradish peroxidase (ABC; 1:100; Vector Labs) in Tris-buffered saline (TBS, pH = 7.4) overnight at 4 °C. The enzyme reaction became visible by using 0.05% 3'3-diaminobenzidine tetrahydrochloride as a chromogen and 0.01% $H_2O_2$ as an oxidant. Sections were post-fixed with 1% $OsO_4$ in 0.1 M PB. Following several washes with distilled water, sections were stained in 1% uranyl acetate, and dehydrated in an ascending series of ethanol. The sections were infiltrated with epoxy resin (Durcupan, Sigma-Aldrich) overnight and embedded on glass slides. After the electrophysiologically recorded cells have been visualized by DAB staining, 3D light microscopic reconstructions were carried out using the Neurolucida system (MBF Bioscience, Williston, VT, USA) with a 100x objective. The length of terminal segments was measured between the terminal tip of the dendrites and the last branching point before the terminal tip. Dendritic spine density on the dendritic branches was calculated as spine/μm between two bifurcations. The percentage of spines on branches was calculated as the percentage of the given spine type from all the types of spines on that branch.

## Acknowledgements

The authors thank Éva Tóth, Katalin Kocsis, Leona Mezei, Otília Kelemen, and Bettina Lehóczki for assistance in anatomical experiments and Gergely Komlósi, Szabina Horváth-Furdan, Ildikó Piszár, Sándor Lovas, Balázs Kovács, Rajmund Lákovics, Márton Rózsa, Gáspár Oláh, Attila Ozsvár, Joanna Grace Sandle and Zoltán Péterfi for electrophysiological recordings.

This work was supported by Eötvös Loránd Research Network grants HUN-REN-SZTE Agykérgi Neuronhálózatok Kutatócsoport and KÖ–36/2021 (GT) Ministry of Human Capacities Hungary (20391-3/2018/FEKUSTRAT and NKP 16–3-VIII-3)(GT); National Research, Development and Innovation Office grants GINOP 2.3.2-15-2016-00018, Élvonal KKP 133807, ÚNKP-20–5 - SZTE-681, 2019–2.1.7-ERA-NET-2022–00038, TKP2021-EGA-09, TKP-2021-EGA-28 (GT) ÚNKP-21–5-SZTE-580 National Institutes of Health awards U01MH114812 (GT) and UM1MH130981 (GT)

## Additional information

### Funding

| Funder | Grant reference number | Author |
|---|---|---|
| Ministry of Human Capacities | 20391-3/2018/FEKUSTRAT | Gábor Tamás |
| National Research, Development and Innovation Office | GINOP 2.3.2-15-2016-00018 | Gábor Tamás |
| National Research, Development and Innovation Office | Élvonal KKP 133807 | Gábor Tamás |
| National Research, Development and Innovation Office | ÚNKP-20-5 - SZTE-681 | Gábor Tamás |
| National Research, Development and Innovation Office | 2019-2.1.7-ERA-NET-2022-00038 | Gábor Tamás |
| National Research, Development and Innovation Office | TKP2021-EGA-09 | Gábor Tamás |
| National Research, Development and Innovation Office | TKP-2021-EGA-28 | Gábor Tamás |

| Funder | Grant reference number | Author |
|---|---|---|
| National Research, Development and Innovation Office | ÚNKP-21-5-SZTE-580 | Gábor Tamás |
| National Institutes of Health | U01MH114812 | Gábor Tamás |
| National Institutes of Health | UM1MH130981 | Gábor Tamás |
| Hungarian Research Network | HUN-REN-SZTE Agykérgi Neuronhálózatok Kutatócsoport | Gábor Tamás |
| Hungarian Research Network | KÖ-36/2021 | Gábor Tamás |

The funders had no role in study design, data collection and interpretation, or the decision to submit the work for publication.

## Author contributions
Pál Barzó, Conceptualization, Investigation, Methodology, Writing – original draft; Ildikó Szöts, Conceptualization, Formal analysis, Investigation, Visualization, Methodology, Writing – original draft, Writing – review and editing; Martin Tóth, Software, Formal analysis, Methodology; Éva Adrienn Csajbók, Data curation, Investigation, Methodology; Gábor Molnár, Conceptualization, Formal analysis, Supervision, Investigation, Visualization, Methodology, Writing – original draft, Writing – review and editing; Gábor Tamás, Conceptualization, Supervision, Methodology, Writing – review and editing

## Author ORCIDs
Gábor Molnár  https://orcid.org/0000-0001-7959-139X
Gábor Tamás  https://orcid.org/0000-0002-7905-6001

## Ethics
Experiments were performed according to the Declaration of Helsinki with the approval of the University of Szeged Ethical Committee and Regional Human Investigation Review Board (ref. 75/2014).

Reviewer #1 (Public review): https://doi.org/10.7554/eLife.100390.3.sa1
Reviewer #2 (Public review): https://doi.org/10.7554/eLife.100390.3.sa2
Reviewer #3 (Public review): https://doi.org/10.7554/eLife.100390.3.sa3
Author response https://doi.org/10.7554/eLife.100390.3.sa4

# Additional files

## Supplementary files
Supplementary file 1. Patients' data for all subjects used in this study.

Supplementary file 2. Number of dendritic spines of the examined pyramidal cells.

MDAR checklist

## Data availability
Source data files have been deposited in Dandiarchive.org under the accession ID: 001281.

The following dataset was generated:

| Author(s) | Year | Dataset title | Dataset URL | Database and Identifier |
|---|---|---|---|---|
| Pál B, Ildikó S, Martin T, Éva Adrienn C, Gábor M, Gábor T | 2025 | Electrophysiology and Morphology of Human Cortical Supragranular Pyramidal Cells in a Wide Age Range | https://dandiarchive.org/dandiset/001281/ | DANDI, 001281 |

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
