## [Editor Report · eLife Assessment]

In this revised work, Barzó et al. assessed the electrophysiological and anatomical properties of a large number of layer 2/3 pyramidal neurons in brain slices of human neocortex across a wide range of ages, from infancy to elderly individuals, using whole-cell patch clamp recordings and anatomical reconstructions. This large data set represents an **important** contribution to our understanding of how these properties change across the human lifespan, supported by **convincing** data and analyses. The authors have addressed the concerns raised in previous reviews. Overall, this study strengthens our understanding of how the neural properties of human cortical neurons change with age and will contribute to building more realistic models of human cortical function.

---

## [Referee Report · Reviewer #1 (Public review)]

Summary:

The manuscript co-authored by Pál Barzó et al is very clear and very well written, demonstrating the electrophysiological and morphological properties of the human cortical layer 2/3 pyramidal cells across a wide age range, from age 1 month to 85 years using whole-cell patch clamp. To my knowledge, this is the first study that look at the cross-age differences biophysical and morphological properties of human cortical pyramidal cells. The community will also appreciate the significant effort involved in recording data from 485 cells, given the challenges associated with collecting data from human tissue. Understanding the electrophysiological properties of individual cells, which are essential for brain function, is crucial for comprehending human cortical circuits. I think this research enhances our knowledge of how biophysical properties change over time in the human cortex. I also think that by building models of human single cells at different ages using these data, we can develop more accurate representations of brain function. This, in turn, provides valuable insights into human cortical circuits and function and helps in predicting changes in biophysical properties in both health and disease.

Strengths:

The strength of this work lies in demonstrating how the electrophysiological and morphological features of human cortical layer 2/3 pyramidal cells change with age, offering crucial insights into brain function throughout life.

Comments on revisions:

Thanks to the authors for addressing my comments and providing greater clarity in the methodology. The analysis is much clearer now. I also appreciate their additional data analysis, particularly on morphology, which strengthens the paper.

---

## [Referee Report · Reviewer #2 (Public review)]

Summary:

In this study, Barzo and colleagues aim to establish an appraisal for the development of basal electrophysiology of human layer 2/3 pyramidal cells across life and compare their morphological features at the same ages.

Strengths:

The authors have generates recordings from an impressive array of patient samples, allowing them to directly compare the same electrophysiological features as a function of age and other biological features. These data are extremely robust and well organised.

The authors group patient ages into developmentally organised bins, which are elaborated on in supplementary analysis - exemplifying the importance of determining early postnatal development on human neuron function

Weaknesses:

The author's use of (perhaps) arbitrary categorisation of spine morphology could limit the full usefulness of these data.

Overall, the authors achieve their aims by assessing the physiological and morphological properties of human L2/3 pyramidal neurons across life. Their findings have extremely important ramifications for our understanding of human brain development and implications for how different neuronal properties may influence life and disease associated with neurological conditions.

Comments on revisions:

Overall, the authors have satisfied my concerns. I fully appreciate their candour with their data and the potential limitations. I especially appreciate their supplementary data inclusions which I believe truly strengthen their conclusions and are a valuable resource for the field,

I agree whole-heartedly with the authors assertion that it is perhaps better to use the most sophisticated equipment, not always being most appropriate. However, statistical rigour should still be standard. As such, my one remaining concern relates to inappropriate replicate choice of spine morphology data in figure 6. I commend the authors inclusion of additional reconstructions and morphology data from further cells in this data set. However, to me, these still represent data from 3 cells and 1 patient/age - as to the best of my interpretation. I feel it would be more helpful to plot cell averages +/- SD for each cell - even if side-by-side with data from all spines. Likewise, it is unclear what statistical test was performed on these data and did it take into account the fact that these values are (a) from 3 technical replicates per group, or (b) that many of the data sets consist of many zero-values (would a categorical test be more appropriate?).

---

## [Referee Report · Reviewer #3 (Public review)]

Summary:

To understand the specificity of age-dependent changes in the human neocortex, this paper investigated the electrophysiological and morphological characteristics of pyramidal cells in a wide age range from infants to the elderly.

The results show that some electrophysiological characteristics change with age, particularly in early childhood. In contrast, the larger morphological structures, such as the spatial extent and branching frequency of dendrites, remained largely stable from infancy to old age. On the other hand, the shape of dendritic spines is considered immature in infancy, i.e., the proportion of mushroom-shaped spines increases with age.

Strengths:

Whole-cell recordings and intracellular staining of pyramidal cells in defined areas of the human neocortex allowed the authors to compare quantitative parameters of electrophysiological and morphological properties between finely divided age groups.

They succeeded in finding symmetrical changes specific to both infants and the elderly, and asymmetrical changes specific to either infants or the elderly. The similarity of pyramidal cell characteristics between areas is unexpected.

Weaknesses:

Human L2/3 pyramidal cells are thought to be heterogeneous, as L2/3 has expanded to a high degree during the evolution from rodents to humans. However, the diversity (subtyping) is not revealed in this paper.

Comments on revisions:

I believe that the current version has been sufficiently revised based on my comments.

---

## [Author Response]

The following is the authors’ response to the original reviews.

**Public Reviews:**

**Reviewer #1 (Public review):**
Summary:The manuscript co-authored by Pál Barzó et al is very clear and very well written, demonstrating the electrophysiological and morphological properties of human cortical layer 2/3 pyramidal cells across a wide age range, from age 1 month to 85 years using whole-cell patch clamp. To my knowledge, this is the first study that looks at the cross-age differences in biophysical and morphological properties of human cortical pyramidal cells. The community will also appreciate the significant effort involved in recording data from 485 cells, given the challenges associated with collecting data from human tissue. Understanding the electrophysiological properties of individual cells, which are essential for brain function, is crucial for comprehending human cortical circuits. I think this research enhances our knowledge of how biophysical properties change over time in the human cortex. I also think that by building models of human single cells at different ages using these data, we can develop more accurate representations of brain function. This, in turn, provides valuable insights into human cortical circuits and function and helps in predicting changes in biophysical properties in both health and disease.Strengths:The strength of this work lies in demonstrating how the electrophysiological and morphological features of human cortical layer 2/3 pyramidal cells change with age, offering crucial insights into brain function throughout life.Weaknesses:One potential weakness of the paper is that the methodology could be clearer, especially in how different cells were used for various electrophysiological measurements and the conditions under which the recordings were made. Clarifying these points would improve the study's rigor and make the results easier to interpret.
**Reviewer #2 (Public review):**
Summary:In this study, Barzo and colleagues aim to establish an appraisal for the development of basal electrophysiology of human layer 2/3 pyramidal cells across life and compare their morphological features at the same ages.Strengths:The authors have generated recordings from an impressive array of patient samples, allowing them to directly compare the same electrophysiological features as a function of age and other biological features. These data are extremely robust and well organised.Weaknesses:The use of spine density and shape characteristics is performed from an extremely limited sample (2 individuals). How reflective these data are of the population is not possible to interpret. Furthermore, these data assume that spines fall into discrete types - which is an increasingly controversial assumption.Many data are shown according to somewhat arbitrary age ranges. It would have been more informative to plot by absolute age, and then perform more rigourous statistics to test age-dependent effects.Overall, the authors achieve their aims by assessing the physiological and morphological properties of human L2/3 pyramidal neurons across life. Their findings have extremely important ramifications for our understanding of human life and implications for how different neuronal properties may influence neurological conditions.
**Reviewer #3 (Public review):**
Summary:To understand the specificity of age-dependent changes in the human neocortex, this paper investigated the electrophysiological and morphological characteristics of pyramidal cells in a wide age range from infants to the elderly.The results show that some electrophysiological characteristics change with age, particularly in early childhood. In contrast, the larger morphological structures, such as the spatial extent and branching frequency of dendrites, remained largely stable from infancy to old age. On the other hand, the shape of dendritic spines is considered immature in infancy, i.e., the proportion of mushroom-shaped spines increases with age.Strengths:Whole-cell recordings and intracellular staining of pyramidal cells in defined areas of the human neocortex allowed the authors to compare quantitative parameters of electrophysiological and morphological properties between finely divided age groups.They succeeded in finding symmetrical changes specific to both infants and the elderly, and asymmetrical changes specific to either infants or the elderly. The similarity of pyramidal cell characteristics between areas is unexpected.Weaknesses:Human L2/3 pyramidal cells are thought to be heterogeneous, as L2/3 has expanded to a high degree during the evolution from rodents to humans. However, the diversity (subtyping) is not revealed in this paper.
**Recommendations for the authors:**

**Reviewer #1 (Recommendations for the authors):**
The manuscript co-authored by Pál Barzó et al is very clear and very well written, demonstrating the electrophysiological and morphological properties of the human cortical layer 2/3 pyramidal cells across a wide age range, from age 1 month to 85 years using whole-cell patch clamp. To my knowledge, this is the first study that looks at the cross-age differences in morphological and electrophysiological properties of human cortical pyramidal cells. The community will also appreciate the significant effort involved in recording data from 485 cells, given the challenges associated with collecting data from human tissue. understanding the electrophysiological properties of individual cells, which are essential for brain function, is crucial for comprehending human cortical circuits. I think this research enhances our knowledge of how biophysical properties change over time in the human cortex. I also think that by building models of human single cells at different ages using these data, we can develop more accurate representations of brain function. This, in turn, provides valuable insights into human cortical circuits and function and helps in predicting changes in biophysical properties in both health and disease.

We are grateful for the positive evaluation of our work. We also thank the reviewers for their comments and believe that our manuscript has improved significantly with their help. In addition to the reviewer’s suggestions for improvement, further cell reconstructions were performed to make the anatomical data more robust (n = 1,2,3,3,4,3,2 additional reconstruction in age groups infant, early childhood, late childhood, adolescence, young adulthood, middle adulthood and late adulthood, respectively; Σn = 18). Four additional cells were added to the spine analysis and the statistics associated with each additional dataset were updated.

I have some comments, particularly regarding the methodology and data presentation, to improve the clarity of the paper(1) I assume the tissue is from the resected area adjacent to the tumor. Could you please clarify this in the Methods section?

Thank you for this comment, it has been clarified in the Methods section with the following sentence: “We used human cortical tissue adjacent to the pathological lesion that had to be surgically removed from patients (n = 63 female n = 45 male) as part of the treatment for tumors, hydrocephalus, apoplexy, cysts, and arteriovenous malformation.”

(2) Regarding the presentation of data in the Methods section, could you please clarify whether the authors used different cells for measuring the various electrophysiological properties? The number of recorded cells for calculating subthreshold properties (e.g., late adulthood: n = 113) differs from the number the cells used for calculating suprathreshold properties (e.g., late adulthood: n = 83). If this is the case, it may make it difficult to compare the electrophysiological properties. Could you please clarify this?

The different element numbers are indeed due to the fact that different quality criteria were defined for the analysis of fast and slow signals. For the analysis of fast signals (e.g. AP half-width, AP upstroke velocity, AP amplitude), higher quality requirements were established therefore cells with high series resistance (> 30 MΩ) were excluded. We have updated and clarified the recording conditions in the text, figures, and methodology section accordingly.

(3) Additionally, they mentioned that their recordings were done at zero holding current and at more than -50 pA. Could you clarify whether the data from these two sets of experiments were combined? If so, please provide an explanation in the methods section.

Basically, we wanted to determine the parameters of the potential changes of the membrane at rest. However, for technical reasons related to the biological amplifier, in some of the experiments a certain continuous holding current may be present during the measurement (3.5% of all experiments). The holding currents were in the range of -50 pA to +60 pA. Within this range, previously checked on mouse neurons we have not found linear correlation between the electrophysiological properties and the holding current. This is reported in the Methods section.

(4) This section needs revision. It is unclear why different series resistances (Rs) or different cells were used to compute various electrophysiological properties." To calculate passive membrane properties (resting membrane potential, input resistance, time constant, and sag) either cells with series resistance (Rs): 22.85 {plus minus} 9.04 MΩ (ranging between -4.55 MΩ and 56.76 MΩ) and 0 pA holding current (n = 154), or cells with holding current > -50 pA (-7.46 {plus minus} 28.56 pA, min: -49.89 pA, max: 59.68pA) and Rs < 30 MΩ (18.96 {plus minus} 6.48 MΩ) (n = 23) were used. For the analysis of high frequency action potential features (AP half-width, AP up-stroke velocity, AP amplitude and rheobase) cells with Rs < 30 MΩ (n = 331 cells with Rs 19.2 {plus minus} 6.6 MΩ) and holding current > -50pA (n = 308 with 0 pA holding current and Rs: 19.22 {plus minus} 6.59 MΩ, n = 23 withholding current: -7.46 {plus minus} 28.56 pA and Rs: 18.96 {plus minus} 6.48 MΩ) were used."

To make the chapter clearer, we simplified the cell groups used to analyse the different electrophysical properties and revised the Method section as follows: “For the analysis of the electrophysiological recordings n = 457 recordings with a series resistance (Rs) of 24.93 ± 11.18 MΩ (max: 63.77 MΩ) were used. For the analysis of fast parameters related to the action potential (AP half-width, AP upstroke velocity, AP amplitude and rheobase), higher quality requirements were set and cells with Rs > 30 MΩ were excluded. This reduced the data set to n = 331 cells with Rs 19.42 ± 6.2 MΩ.”

(5) The authors recorded the sag ratio using a -100 pA injected current. Is there a technical reason why they did not inject more than -100 PA?

There is no particular technical reason, we use similar to others this current amplitude for voltage response recordings over the years to record electrophysiological traces.

(6) In the abstract, the authors mentioned that data were recorded from ages 1 month to 85 years. However, in the results, they stated that data were recorded from ages 0 to 85 years. Could you please clarify this discrepancy?

We corrected this discrepancy.

(7) Additionally, the results mention that data were collected from 485 human cortical layer 2/3 (L2/3) pyramidal cells, but subthreshold membrane features such as resting membrane potential, input resistance, time constant (tau), and sag ratio were calculated in 475 cortical pyramidal cells from 99 patients. Could you please clarify these discrepancies? In the discussion "We recorded from n = 457 human cortical excitatory pyramidal cells from the supragranular layer from birth to 85 years"

Thank you for pointing this out, we have corrected the error. Although our full data set contained 485 pyramidal cells, 28 recordings were excluded from the electrophysiological analysis and were used for morphological evaluation only, therefore 457 recordings were used for passive parameter measurements.

(8) Regarding the distance from the pia to the border layer L1/L2, did the authors notice any differences across ages?

To investigate whether the thickness of cortical layer 1 changes throughout life, we measured the L1 thickness and found no significant differences between age groups (P = 0.09, Kruskal-Wallis test) (Author response image 1).

**Author response image 1. sa4fig1:** Thickness of cortical layer 1 at different life stages. (**A**) Boxplot shows the thickness of layer 1. (**B**) Scatter plot shows the distribution of L1 thickness measured on the reconstructed cells. Age is shown in years on a logarithmic scale, dots are color-coded according to the corresponding age groups.

(9) I am not sure why they referred to the data as layer 2/3 when most of the data, based on Figure 1E, were recorded from a distance of 0-200 µm from the L1/L2 border. Could it be that there is no significant depth-dependent variation in electrophysiological properties, as reported by Berg (2021), Kalmbach (2018), and Chameh (2021)?

Although the vast majority of our data comes from a distance of less than 200 μm from the L1/L2 border, we cannot neglect the fact that our dataset also contains a small number of cells deeper than this, which are layer 3 cells. Apart from some differences shown in Supplementary Figures 7-9, we found no general difference between cells located at a distance of less than 200 μm and more than 200 μm from the L1 border.

(10) In Figure 1, there is variability in resting membrane potential (RMP), tau, and input resistance (IR) within the infant age group. However, this trend is not observed in the sag ratio. Could you please discuss this finding?

The large variance in the data is due to dramatic changes in these three parameters during the first year of life. Supplementary Figure 3 shows the comparisons of parameter distributions of patients between 0-6 months and 6-12 months. The sag amplitude in these cells is generally low therefore no such large changes could have occurred in them.

(11) Did the authors use a K-Nearest Neighbors (KNN) test to assess the accuracy of the infant cluster in Figure 3F?

Based on eight electrophysiological features of the cells (resting Vm, input resistance, tau, sag ratio, rheobase, AP half-width, AP up-stroke, and AP amplitude), the infant pyramidal cells on a UMAP form a distinct group (Author response image 2A) represented by cluster 4 on Author response image 2B. When calculating the sum of the Euclidean distances of cells within the cluster from the centroid, the isolated infant group (cluster 4) shows the smallest distance value from the centroid (cluster 1: 40.2, cluster 2: 36.21, cluster 3: 39.96, cluster 4: 5.72, cluster 5: 39.2, cluster 6: 55.74, cluster 7: 54.27), demonstrating that infant cells create a discrete cluster distinct from other age groups (Author response image 2B).

**Author response image 2. sa4fig2:** (**A**) Uniform Manifold Approximation and Projection (UMAP) of 8 selected electrophysiological properties (resting Vm, input resistance, tau, sag ratio, rheobase, AP half-width, AP up-stroke, and AP amplitude) with data points for 331 cortical L2/3 pyramidal cells, colored with the corresponding age groups. (**B**) UMAP colored by k-means clustering with 7 clusters, red crosses represent the centroids of the clusters.

(12) Missing citation: 'Previous research has shown that the biophysical properties of human pyramidal cells show depth-related correlations throughout L2/3 (Berg et al., 2021).' Please include citations for Kalmbach (2018) and Chameh (2021).

We thank for the additional references, these studies are now cited.

(13) Have they noticed any morphological properties differences among the different cortical lobes (Parietal, Temporal, Frontal, and Occipital). It would be beneficial to present this data, especially since they have a sufficient sample size from each cortical lobe.

The majority of our data set on the morphological properties of pyramidal cells comes from the parietal (n = 17 cells) and temporal lobe (n = 15). We found no significant differences in the morphological properties of cells from these two brain regions and no differences between age groups in the same cortical lobes.

(14) Have the authors found differences in spine characteristics among different cortical areas, as reported previously by 10.1023/a:1024134312173.

We found morphological differences in dendritic spines in the different brain regions, yet, our data are limited to draw definitive conclusions.

**Reviewer #2 (Recommendations for the authors):**
Major(1) I believe that these data presented in all main text figures would be more intuitive to be plotted on a log(age) scale, such as shown in supplementary Figure 13. The bounds of the ages used for different groups, as summarised in Figure 1 feel somewhat arbitrary.

Recent neuroscientific studies on postnatal ageing mainly use the age-group comparison format (Kang 2011, Bethlehem 2022), which has been defined based on milestones in the cognitive, motor, social-emotional, and language/communications domains of observable behaviour (Zubler et al. 2022, for detailed definitions see Kang 2011). Since many parameters do not vary linearly but take a U-shape (or inverted U-shape), statistical quantification of these is not straightforward, so we would retain the age-group format for the main graphs. However, at the reviewer's suggestion, electrophysiological and morphological parameters are presented on a log(age) scale as supplementary figures (Supplementary Figures 2,4 and 6), also further statistical analysis was also carried out without grouping the data (see response 5).

(2) The authors present a lot of data values in the text, which is also shown in the figures. This makes reading of the manuscript somewhat difficult in places. For brevity, it may be best to present this data as supplementary tables.

Thank you for this suggestion. We have inserted these data as tables.

(3) I am unclear why the authors excluded cells that fired doublets or triplets in Figure 4? Were these included in the passive and AP-specific analysis - but excluded from F-I plots? Please clarify the rationale and the relative abundance of these physiological types based on age - one might predict that more initial-burst firing types are associated with older neurons?

Thank you for drawing attention to this anomaly. We have updated the figures and text by adding the cells with initial burst firing. These cells are also included in the analysis of passive and action potential properties. In our overall dataset, 6.78% of cells show burst firing; infant: 0%, early childhood: 3.57% (1 cell), late childhood: 0%, adolescence: 11.11% (6 cells), young adulthood: 10.11% (9), middle adulthood: 10.71% (6 cells), late adulthood: 7.96 (9 cells) of all cells including the age groups.

(4) The statistical analyses performed in Figure 6 are not justified. From the authors' description of these data, they derive spine density measurements from 1 infant and 1 aged adult, then perform pseudoreplicated analysis in these individuals. These data would require greater replication from infant and aged groups - with the possible inclusion of a younger adult group also. It would be ideal to have n=3/age group to allow robust statistical analysis.

Thank you for this point. Accordingly, we have expanded our data set to include n = 3 infant pyramidal cells (83 days old, from one patient) and n = 3 pyramidal cells from three late adulthood patients (64.3 ± 2.08 years old).

(5) Given the high number of individuals and replicates throughout this manuscript, a more circumspect approach to statistics would be appreciated, e.g. a generalised linear mixed effects model - with age as a fixed effect and sex, patient, etc as random effects. This may reveal the greatest statistical power of these important and rich data.

Of the generative models we used the Generalized Additive Mixed Model (GAMM) to describe the relationship between age and the various passive and active electrophysiological features. We defined age with cubic spline smoothing term as the fixed effect and gender, brain area, surgical procedure, and hemisphere as random effects. With GAMM we found that the age-dependent correlation of the examined parameters (resting membrane potential, input resistance, tau, sag ratio, rheobase current, AP half-width, AP up-stroke velocity, AP amplitude, first AP latency, adaptation) was significant, except for F-I slope, described by the model incorporating the four random effects. We also observed correlation with gender, brain area, hemisphere, and surgical procedure in various intrinsic properties. The Author response table 1 below shows the statistical values of GAMM and the statistical tests used in the manuscript to compare.

**Author response table 1. sa4table1:** 

Electrophysiological parameters	Age P-value	Age group P-value	Gender P-value	Gender (all age groups)	Brain area P-value	Hemisphere P-value	Surgical procedure P-value	Surgical procedure (all age groups)
Statistical analysis	GAMM	Kruskal-Wallis	GAMM	Pairwise*	GAMM	GAMM	GAMM	Pairwise** (tumor-shunt)
Resting Vm (mV)			0.95	0.68		0.0498	0.025	0.11
Input resistance (M)			0.75	0.2				0.02
Tau (ms)			0.44	0.03		0.09		0.28
Sag ratio			0.44	0.003	0.09			0.34
Rheobase (pA)			0.68	0.13	0.016	0.4	0.87	
AP half-width (ms)						0.31	0.61	
AP up-stroke (mV/ms)			0.83	0.03		0.02	0.012	0.03
AP amplitude (mV)	<2*10–16		0.42	0.86		0.11	0.74	0.14
F-I slope	0.055	0.055	0.97	0.6	0.02	0.07	0.7	0.6
First AP latency (ms)	0.01		0.59	0.17	0.15	0.89	0.8	0.71
Adaptation	0.03	0.032	0.46	0.95	0.002	0.28	0.34	0.88

Statistical significance of patient attributes *In the pairwise comparison, the age of cells in the two groups was significantly different: female (subthreshold: 37.36 ± 26.25 years old, suprathreshold: 38.3 ± 25.6 y.o.) - male (subthreshold: 24.86 ± 23.7 y.o., suprathreshold: 25.7 ± 23.93 y.o.), subthreshold: P = 1.96*10-6, suprathreshold: P = 3.25*10-5 Mann-Whitney test.

**In the pairwise comparison, the age of cells in the two groups was significantly different: surgical procedure: tumor removal (subthreshold: 33.72 ± 24.33 y.o., suprathreshold: 36.43 ± 27.07 y.o.) - VP shunt (subthreshold: 27.38 ± 29.69 y.o., suprathreshold: 27.07 ± 29.37 y.o.) subthreshold: P = 3.68*10-3, suprathreshold: P = 1.64-10-3, Mann-Whitney test

(6) Regarding the morphological diversity of dendritic spines. There is some debate in the field as to whether the distinction of specific dendritic spine types - as conveyed in this manuscript - are true subtypes or reflect a continuum of diverse morphology (see Tønneson et al., 2014 Nature Neuroscience). It is appreciated that the approach taken by the authors is the dogma within the field - however, dogma should continue to be challenged. Given that the authors have used DAB labelling combined with light microscopy, the possibility of accurately measuring spine morphology required for determining this continuum is extremely limited (e.g. Li et al., (2023) ACS Chemical Neuroscience). I would suggest that alongside the inclusion of further replicates for their spine analysis, the authors tone down their discussion of spine subtypes given the absence of any synaptic data presented in this current study to support the maturation (or otherwise) of dendritic spine synapses.

Many thanks to the reviewer for this comment. We agree with the drawbacks of our method for testing spine categorization. To increase the reliability of our results, we increased the number of pyramidal cells in the infant and late adult groups. We also revised the figure and as suggested by Reviewer#3 added photos of spines to each category in addition to schematic drawings to give an impression of the phenotype. In the discussion, we only address the differences between two readily separable mushroom and filopodial forms and highlight results that only confirm findings already known in the literature. Although the concerns are valid, we apply the sentence from the above Li et al. (2023) reference “...the most sophisticated equipment may not always be necessary for answering some research questions”. We believe that it is worth sharing our data and the somewhat subjective grouping, which we hope to report in more detail in the future.

Minor(1) The order of the supplemental materials is out of order with their introduction in the text. These should be revised to reflect the order mentioned in the text.

Thank you for your comment, we have corrected the order of the supplementary figures.

(2) In Supplementary Figure 13, it would be informative to include some form of linear regression to confirm whether an age-dependent effect on neuronal morphology exists.

We have added linear regression to the figure.

(3) Figure 3D = should this be AP - not Ap?

Thank you for drawing attention to this, we have corrected the incorrect typing on the figure.

(4) For UMAP analysis in Figure 3, please provide a table of the features that were used for the 32 & 8-parameter UMAPs respectively.

We have added a table to the Materials and methods section of all the electrophysiological features included in the UMAP.

(5) For morphology, please include pia and L1/2 border for reconstructions shown for clarity.

We indicated both the pia mater and the L1/2 border on the figure showing all the reconstructions (Supplementary Figure 10).

**Reviewer #3 (Recommendations for the authors):**
Major:(1) Data were obtained from different cortical areas of human patients of different ages. The electrophysiological characteristics were largely independent of other attributes such as disease, gender, and cortical areas (Supplementary Figure 2). To support the conclusion that age is one of the key attributes responsible for change, a similar morphological analysis would be necessary for gender.

We updated the text and the supplementary section with Supplementary Figures 18-21. to determine if age-related differences in biophysical characteristics are affected by the patient's gender.

(2) 'mushroom-shaped, thin, filopodial, branched, and stubby spines'

Show photographs of individual typical spine types to make the classification easier to understand.

To make the classification more understandable, we have updated the corresponding figure (Figure 6) with representative photos of the dendritic spine types.

(3) Some electrophysiological parameters of the infant group showed higher deviations compared to other age groups. A UMAP (Supplementary Figure 2) shows that some infant neurons form a small cluster, while other infant neurons are scattered with neurons of other ages. Are there any differences between infant neurons in the small cluster and other infant neurons with respect to attributes other than age?

For most of the electrophysiological parameters, the infant age group showed age-dependent variability, as illustrated in Supplementary Figures 3, 2,4 and 6 . The small group of infant cells is not clustered by gender, brain region, or medical condition, as shown in Supplementary Figure 5.

(4) A recent paper (Benavides-Piccione et al. 2024, doi:10.1093/cercor/bhae180) reported that some morphological parameters of human layer 3 neurons differ between occipital and temporal regions. Area-dependent morphological differences have been also reported in non-human primates. Discussion of potential contradictions may therefore be requested.

Most of the cells we reconstructed originated from the parietal and temporal regions (parietal: n = 20, temporal: n = 23, frontal: n = 15, occipital: n = 5). We found no differences in morphological features between these two regions, and we also found no significant differences when we compared the cells from the same brain regions by age group.

(5) L2/3 cells of rodents are morphologically differentiated according to cortical depth. If individual L2/3 cells of humans are less differentiated than those of rodents, this point should be discussed.

Depth-related morphological heterogeneity has already been reported previously (Berg 2021), however, our dataset on the morphological characteristics of pyramidal cells is from the upper L2/3 region, with their soma located at a distance of 117.85 ± 65.3 μm (between: 11.05 and 243.3 μm) from the L1/L2 border. Therefore, we cannot conclude from our data whether humans are less differentiated than rodents.

Minor:(1) Cell body morphology may affect electrophysiological properties. However, morphological quantification of cell bodies has not been reported. It may be added.

In our DAB-labeled samples, we could not perfectly measure the total volume of the cell body in the reconstructions, therefore our measurements regarding the soma morphology are not shown in the manuscript. When comparing the cell body area of the middle sections of the soma of the reconstructed cells between the age groups, we found no significant differences (P = 0.082, Kruskal–Wallis test).

(2) 'The adaptation of the AP frequency response'Describe how this parameter was obtained.

The adaptation of the AP frequency response or adaptation was calculated as the average adaptation of the interspike interval between consecutive APs.

(3) 'we excluded cells showing initial duplet or triplet action potential bursts'Why were the burst cells excluded from the analysis?

We have modified the figures and text to include cells with initial burst firing.

(4) Electrophysiological characteristics to be analyzed:Spike thresholds and afterhyperpolarizations

We found age-related differences in the amplitude of the afterhyperpolarization (P = 2.56*10^-30^, Kruskal-Wallis test) and in the threshold of the action potential (P = 5.24*10^-12^, Kruskal-Wallis test) (Author response image 3).

**Author response image 3. sa4fig3:** Age-dependence of afterhyperpolarization and AP threshold. (**A-B**) Boxplots show the differences in afterhyperpolarization (AHP) amplitude (A) and AP threshold (B) between age groups. Asterisks indicate statistical significance (* P < 0.05, ** P < 0.01, *** P < 0.001, Kruskal-Wallis test with post-hoc Dunn test). (**C-D**) Scatter plots show AHP amplitude (C) and AP threshold (D) across the lifespan. Age is shown on a logarithmic scale, dots are colored according to the corresponding age group.

(5) 'We identified and labeled each spine on n = 2 fully 3D-reconstructed cells'To which cortical area do these cells belong?At what depths are they distributed?Is it possible to report the number of spines, in addition to the density per unit length?

We increased the number of cells in which we analyzed dendritic spine density. The data shown in Figure 6. are from pyramidal cells from an infant patient (n = 3 from a single patient) and late adulthood patients (n = 3 from 3 patients) (Supplementary Figure 13). The infant cells are from the same patient, the sample is from the right parietal lobe, and the patient is 83 days old. The older cells are from three different patients (#1: 65 years old, right temporal lobe; #2: 66 years old, right parietal lobe; #3: 62 years old, right frontal lobe). Infant cells are located 144.43 ± 45.26 µm (#1: 109.3, #2: 128.49, #3: 195.5 µm), late adult cells 161.22 ± 66.22 µm (#1: 183.5, #2: 213.42, #3: 86.73 µm) from the L1/2 border. We provide the number of spines in an additional supplementary table (Supplementary table 2.).